# MAD: Manifold Attracted Diffusion

## Abstract

Score-based diffusion models are a highly effective method for generating samples from a distribution of images. We consider scenarios where the training data comes from a noisy version of the target distribution, and present an efficiently implementable modification of the inference procedure to generate noiseless samples. Our approach is motivated by the manifold hypothesis, according to which meaningful data is concentrated around some low-dimensional manifold of a high-dimensional ambient space. The central idea is that noise manifests as low magnitude variation in off-manifold directions in contrast to the relevant variation of the desired distribution which is mostly confined to on-manifold directions. We introduce the notion of an extended score and show that, in a simplified setting, it can be used to reduce small variations to zero, while leaving large variations mostly unchanged. We describe how its approximation can be computed efficiently from an approximation to the standard score and demonstrate its efficacy on toy problems, synthetic data, and real data.

## 1 Introduction

Score-based diffusion models are a state-of-the-art approach for image synthesis, often outperforming alternatives like generative adversarial networks (GANs) and variational autoencoders (VAEs) in sample quality and diversity (Ho et al., 2020; Song & Ermon, 2020). These models learn a neural network by adding noise to data samples during training, according to some forward process. The network can then be used to reverse this process during inference. Essentially, they denoise an image with independent Gaussian pixel values into samples from the data distribution.

However, real-world datasets are often corrupted by noise arising from measurement errors, compression artifacts, or data collection processes (Gupta & Gupta, 2019; Brummer & De Vleeschouwer, 2019). In this case, standard diffusion models trained directly on noisy data will learn to reproduce this corruption in their generated samples. We aim to address this by developing a method that can generate samples that approximately come from the clean distribution, despite being trained on noisy data.

A well-established paradigm for understanding data with a high-dimensional representation is the *manifold hypothesis*: meaningful data distributions are concentrated near a low-dimensional manifold embedded in the ambient space (Bengio et al., 2013; Fefferman et al., 2016). For instance, natural images, despite being represented as a $d$-dimensional array, where $d$ can be in the millions, exhibit intrinsic dimensions that are far lower (Pope et al., 2021). This viewpoint will be the basis of our approach as we interpret noise as low-magnitude variations in directions orthogonal to the manifold, whereas meaningful variations of the underlying data correspond to movements along the manifold itself. The central idea is to exploit this geometric structure during the inference process to suppress the former while preserving the latter.

The manifold hypothesis has previously been considered in the context of diffusion models to study their convergence behavior, e.g. in Bortoli (2022), Tang & Yang (2024), and Potaptchik et al. (2024). It has also been argued in Stanczuk et al. (2024) that diffusion models can be leveraged to estimate the intrinsic dimension of data manifolds. Some existing works tackle the problem of noisy training (Daras et al., 2023; Lu et al., 2025), but require adapting the training process. Outside the generative setting, there are also traditional manifold denoising techniques predating diffusion models (Hein & Maier, 2006; Gong et al., 2010; Wang & Carreira-Perpinán, 2010; Fefferman et al., 2018; Faigenbaum-Golovin & Levin, 2023).

This work introduces *Manifold Attracted Diffusion (MAD)*, an efficiently implementable modification to the inference procedure of score-based models. We define the concept of an "extended score", which coincides with the standard score when that exists, but is also well-defined for Dirac delta distributions. In fact, it treats them essentially as Gaussians with a certain non-zero variance. This property can be leveraged in the inference procedure to reduce small off-manifold variations to almost zero while leaving larger on-manifold variations mostly unchanged, resulting in a soft thresholding effect. Thereby our method implicitly "attracts" the generated samples towards a low-dimensional structure, effectively filtering out noise. Importantly, a suitable approximation to the extended score can be computed easily from an approximation to the standard score, which enables the use of established training methods as well as pretrained networks under some conditions that are, e.g., satisfied by the framework of Karras et al. (2022). Our key contributions include:

- The formal definition and analysis of the extended score;

- An inference algorithm that reduces noise in generated samples without needing a special training procedure, making it compatible with established frameworks and pretrained models;

- Empirical validation on toy problems and real-world image data, such as FFHQ, AFHQ, ImageNet, and EMPIAR-11618 (cryo electron microscopy data).

We also distinguish our work from other diffusion-based approaches. Methods for posterior sampling, such as DPS (Chung et al., 2023; 2022), are designed to solve inverse problems (Daras et al., 2024), such as denoising a single given image. Our goal is different: we learn from a dataset of noisy images to produce new, clean samples from the underlying distribution.

This paper is structured as follows. In Section 2, we review the necessary background on score-based diffusion models. In Section 3, we formally introduce the extended score and analyze its properties. Section 4 details our proposed inference algorithm and, in Section 5, we present numerical experiments that validate our approach.

## 2 BACKGROUND

We will be working with the probability flow ODE formulation of diffusion models, largely following the framework of Karras et al. (2022). For more background and the connection to other diffusion model formulations we refer to Karras et al. (2022) and the references therein. Given a data distribution $p_0$ on $\mathbb{R}^d$ we consider the stochastic process $(X_\sigma)_{\sigma \in [0, \sigma_{\max}]}$ with $X_0 \sim p_0$ and

$$X_\sigma = X_0 + \mathcal{N}(0, \sigma^2 \mathbb{I}), \quad \sigma > 0.$$

The corresponding densities are given by

$$p_\sigma = p_0 * g_{\sigma^2}, \tag{1}$$

where $g_{\sigma^2}(x) = (2\pi\sigma^2)^{-d/2} \exp(-\frac{\|x\|^2}{2\sigma^2})$ is the density of $\mathcal{N}(0, \sigma^2 \mathbb{I})$. The central idea behind score-based diffusion models is to generate a sample $x_0 \sim X_0$ from a sample $x_{\sigma_{\max}} \sim X_{\sigma_{\max}}$, where $x_{\sigma_{\max}}$ is, in practice, approximated by simply sampling from $\mathcal{N}(0, \sigma_{\max}^2 \mathbb{I})$, as for a sufficiently large $\sigma_{\max}$ this should only introduce a negligible error. One way to achieve this is by evolving the ODE

$$\mathrm{d}x_t = -\dot{\sigma}(t)\sigma(t)Sp_{\sigma(t)}(x_t)\mathrm{d}t, \tag{2}$$

where $\sigma \colon [0, T] \to [0, \sigma_{\max}]$ is some noise schedule and the score operator is given by

$$S \colon P(\mathbb{R}^d) \to C(\mathbb{R}^d, \mathbb{R}^d), \qquad p \mapsto \nabla_x \log p,$$

where $P(\mathbb{R}^d) := \{p \in C^1(\mathbb{R}^d, \mathbb{R}) \colon \int_{\mathbb{R}^d} p(x)\,dx = 1, \ p(x) > 0 \ \forall x \in \mathbb{R}^d\}$ is the set of densities that are positive everywhere in $\mathbb{R}^d$. Note that, for $\sigma > 0$, $p_0 * g_{\sigma^2}$ is a density even if $p_0$ is not. The fact that the score may not be well-defined for $p_0$, e.g. because the data distribution is supported on some lower-dimensional subset in $\mathbb{R}^d$, is avoided in practice by generating a sample $x_\delta$ with $\delta$ close to 0. It can be shown (Song et al., 2021; Karras et al., 2022) that evolving a sample $x_{t_1} \sim X_{t_1}$ from $t_1$ to $t_2$ according to (2) yields a sample $x_{t_2} \sim X_{t_2}$.

Of course, this is only useful provided that we have access to the score, which depends on the data distribution, and from which we usually have only a finite number of samples. Remarkably, it turns

out that a useful approximation of the score can be learned by training a neural network on these samples. A common practice, motivated by numerical stability, is to learn the so-called denoiser function $D$, which is simply a shifted and scaled version of the score, i.e. $D(x, \sigma) = \sigma^2 Sp_\sigma(x) + x$. This denoiser function is then approximated by a neural network $D_\theta$ using a loss based on

$$\mathbb{E}_{y \sim X_0} \mathbb{E}_{\eta \sim \mathcal{N}(0, \sigma^2 \mathbb{I})} \|D_\theta(y + \eta) - y\|_2^2$$

which is minimized by the denoiser function. However, as the expectation over the data distribution must be replaced by the empirical expectation based on the available data samples, this loss has, in general, many global minima. Nonetheless, employing neural networks utilizing an adapted U-Net architecture (Ronneberger et al., 2015; Ho et al., 2020; Dhariwal & Nichol, 2021) appears to introduce sufficient bias towards a good approximation of the score. This approximation capability is primarily founded on the empirical observation that using it to generate images by evolving equation 2 (or related differential equations) produces realistic samples. An analytical description of how $D_\theta$ approximates $D$ is still an open problem. As the purpose of the work is to introduce a novel inference procedure, we will simply assume that we can obtain a suitable approximation to the score by some established training method.

## 3 CONCEPT: THE EXTENDED SCORE

Let $M(\mathbb{R}^d)$ denote the set of probability measures on $\mathbb{R}^d$.

**Definition 3.1** (Extended score). *For $d \in \mathbb{R}^d$, $p \in M(\mathbb{R}^d)$, $\gamma \in (0, \infty)$, let*

$$H_\gamma \colon M(\mathbb{R}^d) \to C(\mathbb{R}^d, \mathbb{R}^d), \qquad p \mapsto (1 + \gamma)S(p * g_\gamma) + \gamma \tfrac{\mathrm{d}}{\mathrm{d}\gamma} S(p * g_\gamma)$$

*and*

$$H_0 \colon \widetilde{M}(\mathbb{R}^d) \to \{f \colon \mathbb{R}^d \to \mathbb{R}^d\}, \qquad p \mapsto \lim_{\gamma \to 0} H_\gamma p,$$

*where $\widetilde{M}(\mathbb{R}^d) := \{p \in M(\mathbb{R}^d) \colon \lim_{\gamma \to 0}(H_\gamma p)(x) \in \mathbb{R}^d \ \forall x \in \mathbb{R}^d\}$.*

We first note that $H_0$ coincides with the score for probability distributions with density in $P(\mathbb{R}^d)$, which we can view as a subset of $M(\mathbb{R}^d)$ by identifying a density function $p \in P(\mathbb{R}^d)$ with the measure given by $p(A) = \int_A p(x)\mathrm{d}x$ for $A \subseteq \mathbb{R}^d$ (see Section A.2 for a proof).

**Lemma 3.2.** *Let $p \in P(\mathbb{R}^d)$. Then $H_0 p = Sp$.*

It is, however, also well-defined for, e.g., Dirac delta measures. Specifically, let $\delta$ denote the Dirac delta at 0, then

$$H_\gamma \delta(x) = (1 + \gamma)S(g_\gamma)(x) + \gamma \tfrac{\mathrm{d}}{\mathrm{d}\gamma} S(g_\gamma)(x) = -\tfrac{(1+\gamma)x}{\gamma} + \gamma \tfrac{\mathrm{d}}{\mathrm{d}\gamma}\left(-\tfrac{x}{\gamma}\right) = -x.$$

In particular it holds that $H_0 \delta(x) = -x$, i.e. we obtain[1] a function which, similar to the score for Gaussians, yields at each point a vector pointing towards the mode of the probability distribution. This generalizes to mixtures of Dirac delta distributions, where $H_0 p$ will point towards the location of the nearest Dirac delta in the mixture (see Section A.2 for a proof).

**Lemma 3.3.** *Let $n \in \mathbb{N}$, $\mu_1, \ldots, \mu_n \in \mathbb{R}^d$, $c_1, \ldots, c_n \in \mathbb{R}_+$ such that $\sum_i c_i = 1$ and $p = \sum_{i \in [n]} c_i \delta_{\mu_i}$. Then*

$$H_0 p(x) = -\sum_{i \in [n]} z_i(x)(x - \mu_i),$$

*where $W_i = \{x \in \mathbb{R}^d \colon \|x - \mu_i\| \leq \|x - \mu_j\| \ \forall j \in [n]\}$ is a Voronoi region and*

$$z_i(x) = \begin{cases} 0 & x \notin W_i, \\ 1 & x \in \operatorname{int} W_i, \\ c_i \left(\sum_{j \colon x \in W_j} c_j\right)^{-1} & x \in \partial W_i. \end{cases}$$

---

[1]Note that in this simple case, $H_\gamma \delta$ is already the same as $H_0 \delta$, which is not the case in more complicated scenarios, e.g. for mixtures of Dirac deltas.

Note that the first two cases in the expression for $z_i(x)$ cover almost every (w.r.t the Lebesgue measure) $x \in \mathbb{R}^d$ and the third case is only needed if $x$ is equally distant to multiple $\mu_i$.

By combining the expressions of the extended score for a Dirac delta and for a non-degenerate Gaussian distribution, it is possible to derive an explicit expression of the extended score for any (possibly degenerate) Gaussian distribution. This is achieved by exploiting how the extended score behaves with respect to products of measures, see Appendix A.1 for the details.

We have the rather peculiar property that $H_0\delta = Sg_1$, i.e. the extended score of the Dirac delta matches the score of a variance 1 Gaussian. At this point, it should be noted that it is impossible to find a nice extension of the score operator that includes Dirac delta distributions, as

$$Sg_\gamma(x) = -\tfrac{x}{\gamma},$$

which diverges for $x \neq 0$ as $\gamma \to 0$. As such any extension is necessarily discontinuous with respect to any topology in which $\lim_{\gamma \to 0} g_\gamma = \delta$.

However, the ability of the extended score to treat distributions with positive variance like they were Dirac delta distributions will, in fact, be the cornerstone of our proposed inference technique. Under the manifold hypothesis, the clean data distribution has significant variation only along a small number of directions. Given noise in the ambient pixel space, which has a much higher dimension than the image manifold, the variance in off-manifold directions due to noise should be much smaller (e.g. for isotropic Gaussian noise $\eta$ the variance in each direction is of order $\sim d^{-\frac{1}{2}}\mathbb{E}[\|\eta\|]$). Thus, noise can be suppressed by using the extended score. This principle is formally justified in Appendix A.1, where we derive the extended score for a distribution supported on a low-dimensional subspace and show that it strongly attracts samples in the off-manifold directions while preserving the standard score in the on-manifold directions.

## 4 IMPLEMENTATION

In view of the property of the extended score just discussed, we would like to design an inference procedure that is able to generate samples with less noise, if compared to the samples obtained via the usual score. To this end we first note that, due to (1), we have

$$S(p_{\sigma(t)} * g_\gamma) = S(p_0 * g_{\sigma(t)^2} * g_\gamma) = S(p_0 * g_{\sigma(t)^2+\gamma}) = S(p_{\sqrt{\sigma(t)^2+\gamma}}).$$

So, given a network trained to approximate the score for any $\sigma \in [0, \sigma_{\max}]$, as is the case in the framework of Karras et al. (2022), we also have an approximation to the extended score with a given small $\gamma > 0$, where the $\gamma$-derivative can be obtained, e.g., by a finite difference[2] approximation. As the desirable properties of the extended score hold in the $\gamma \to 0$ limit, which we cannot compute directly, we instead need to choose a suitable dependence $\gamma(t)$ with $\lim_{t \to 0} \gamma(t) = 0$.

A basic way to conduct inference with the standard score is evolving (2) simply via Euler method, i.e. initializing with $x_0 \sim \mathcal{N}(0, \sigma(t_0)^2\mathbb{I})$ and iterating

$$x_{i+1} = x_i - (t_{i+1} - t_i)\dot{\sigma}(t_i)\sigma(t_i)S_\theta(\sigma(t_i), x_i), \tag{3}$$

where $S_\theta(\sigma, x) \approx Sp_\sigma(x)$ is the learned approximation of the score. We will instead, based on Definition 3.1, consider the iteration

$$x_{i+1} = x_i - m(t_i)(t_{i+1} - t_i)\dot{\sigma}_\gamma(t_i)\sigma_\gamma(t_i)\big((1+\gamma(t_i))S_\theta(\sigma_\gamma(t_i), x_i) + \gamma(t_i)\tfrac{\mathrm{d}}{\mathrm{d}\gamma}S_\theta(\sigma_\gamma(t_i), x_i)\big), \tag{4}$$

where $\sigma_\gamma(t) = \sqrt{a\sigma(t)^2 + b\gamma(t)}$, $a, b > 0$ are manually chosen parameters, and $m(t_i)$ is a correction factor determined as explained below. Note that this reduces to (3) for $a = 1$, $b = 0$, and $m(t) = (1+\gamma(t))^{-1}$, so essentially the choices of $a$, $b$, and $m$ determine to what extent we would like the inference to push points onto a manifold at a given time step. More precisely, increasing $a$ or reducing $b$ corresponds to lessening the effect of the extended score and therby limiting the denoising effect. Conversely, reducing $a$ or increasing $b$ yield a stronger denoising effect. As such, $a$ and $b$ can be seen as regularization parameters, and their choice is problem dependent. While in

---

[2]Note that one could also compute the exact derivative of the score network using automatic differentiation (AD). However, since PyTorch is not optimized for forward mode AD this is significantly more expensive than a second evaluation of the network and did not yield any clear improvements in the numerical experiments.

the idealized case of the $\gamma \to 0$ limit, analyzed in Section 3, Diracs and variance 1 Gaussians are treated in the same way, the choice of $a$ and $b$ adjusts this equivalence in practice, causing variations below this (soft) threshold to be compressed significantly. Note that we currently do not determine the explicit dependence of this threshold on the parameters. However, since we cannot expect to know the optimal threshold for a data distribution in practice, a parameter optimization seems unavoidable anyway. We further enforce $\sigma_\gamma(t) = t$ to match the scheduling of Karras et al. (2022), as it allows us to take advantage of their optimized choice of time steps and leads to better comparability. Lastly, we introduce a parameter $p > 0$ to regulate the relative speed of convergence of $\sigma$ and $\gamma$ via $\gamma(t) = \sigma(t)^p$.

With these choices, (4) simplifies to

$$x_{i+1} = x_i - m(t_i)(t_{i+1} - t_i)t_i\big((1 + \gamma(t_i))S_\theta(t_i, x_i) + \tfrac{b\gamma(t_i)}{2t_i}\tfrac{\mathrm{d}}{\mathrm{d}\sigma}S_\theta(t_i, x_i)\big), \quad (5)$$

where $\frac{\mathrm{d}}{\mathrm{d}\sigma}S_\theta$ denotes the derivative of $S_\theta \colon [0, \sigma_{\max}] \times \mathbb{R}^d \to \mathbb{R}^d$ w.r.t its first argument, and $\gamma(t_i)$ is a solution of

$$\sqrt{a\gamma(t_i)^{\frac{2}{p}} + b\gamma(t_i)} = t_i.$$

We observe that in the case of $S_\theta(\sigma, x) = -\frac{x-\mu}{\sigma^2}$, which corresponds to an initial distribution consisting of a Dirac delta at $\mu$, the standard score inference rule (3) with $\sigma(t) = t$ evaluates to

$$x_{i+1} = x_i - \tfrac{t_i - t_{i+1}}{t_i}(x_i - \mu) = (1 - \tfrac{t_i - t_{i+1}}{t_i})x_i + \tfrac{t_i - t_{i+1}}{t_i}\mu. \quad (6)$$

This means that, if $\frac{t_i - t_{i+1}}{t_i} \geq \Delta > 0$ for every $i$, we have

$$\|x_{i+k} - \mu\| \leq (1 - \Delta)^k\|x_i - \mu\|,$$

i.e. convergence to $\mu$ at a geometric rate. We would like our inference rule to mimic this behavior for simple Dirac deltas, which is accomplished by choosing the correction factor as

$$m(t_i) = \Big(1 + \gamma(t_i) - \tfrac{b\gamma(t_i)}{t_i^2}\Big)^{-1}, \quad (7)$$

see Section A.2 in the appendix for a detailed derivation. Note that, as long as $\gamma(t_i) \in o(t_i^{-2})$ we have $\lim_{t\to 0} m(t) = 1$. Putting it all together, we arrive at the following algorithm.

---

**Algorithm 1** Inference with extended score

1: **function** INFERENCE($S_\theta, a, b, p, \delta, (t_i)_{i\in\{0,\dots,N\}}$)
2:     **sample** $x_0 \sim \mathcal{N}(0, t_0^2\mathbb{I})$                                   ▷ Generate Gaussian sample
3:     **for** $i \in \{0, \dots, N-1\}$ **do**
4:         $\gamma_i = \textbf{solve}(a\gamma_i^{2/p} + b\gamma_i - t_i^2 = 0)$      ▷ Determine $\gamma_i$ using a root finding algorithm
5:         $s_i = S_\theta(t_i, x_i)$                                  ▷ Evaluate the score network
6:         $\widetilde{s}_i = S_\theta((1 + \delta)t_i, x_i)$
7:         $s_i' = \tfrac{\widetilde{s}_i - s_i}{\delta t_i}$                         ▷ Compute an approximation to the derivative
8:         $m_i = (1 + \gamma_i - \tfrac{b\gamma_i}{t_i^2})^{-1}$                  ▷ Compute correction factor
9:         $x_{i+1} = x_i - m_i(t_{i+1} - t_i)t_i((1 + \gamma_i)s_i + \tfrac{b\gamma_i}{2t_i}s_i')$      ▷ Update sample
10:     **return** $x_N$

---

## 5 NUMERICAL EXPERIMENTS

In section 5.1 (toy examples in $\mathbb{R}^2$) and in section 5.2 (FFHQ, AFHQv2, and ImageNet) we present some illustrative numerical simulations to build intuition of the "manifold attraction" property of the extended score on clean datasets. Further, in sections 5.3 (synthetic dataset) and 5.4 (CIFAR-10) we provide controlled experiments showing qualitatively and quantitatively (FID scores) the denoising effect of the extended score. Finally, in section 5.5 we test MAD with real Cryo-EM data.

We emphasize that MAD addresses a specific blind generative denoising task where only noisy data is available, and the specific degradation model may be unknown. In this context, there are no established benchmarks, and a comparison with other (non-generative) manifold denoising approaches would not be meaningful, because they tackle a different problem.

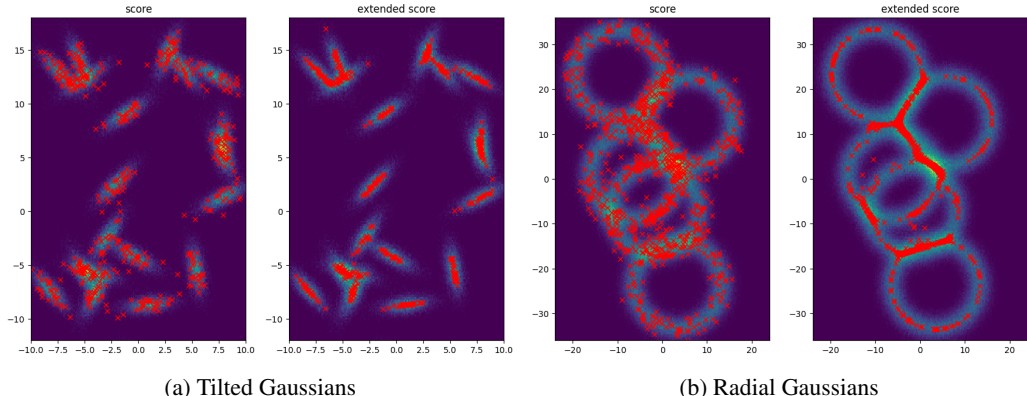

(a) Tilted Gaussians          (b) Radial Gaussians

Figure 2: Target distribution is displayed as a histogram colormap. Red crosses indicate samples generated either via standard score inference (left) or extended score inference (right). Extended score parameters are $p = 1.3$, $a = 1$, $b = 1.1$ for (a) and $p = 2$, $a = 2$, $b = 30$ for (b).

## 5.1 ILLUSTRATIVE EXAMPLES IN $\mathbb{R}^2$

We visualize the evolution according to the extended score in $\mathbb{R}^2$ for relatively simple distributions, where the (extended) score can be computed explicitly (see Appendix A.1). Figure 1 displays the inference trajectories for a degenerate Gaussian mixture, which behaves like a Dirac delta at $0$ in the $x_2$-direction and a Gaussian in the $x_1$-direction as the means are chosen such that the different mixture components affect each other very little. We can see that for the Gaussians with variances $0.2$ and $0.5$ respectively, using the extended score causes all trajectories to end up at the respective means, i.e. it behaves as if we had Dirac deltas (in the $x_1$-direction) at these locations. For the higher variance Gaussians the extended score trajectories still end up closer to the mean than they would for standard score, but this variance reduction effect decreases significantly as the variance of the Gaussians increases. All together, we essentially have a soft thresholding effect, where variances below a certain value are shrunk to $0$, while large enough variances are left almost unchanged.

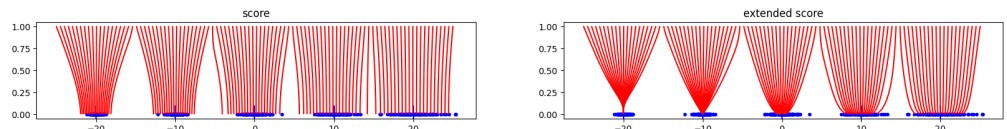

Figure 1: Comparison of inference trajectories (red) and samples (blue) for Gaussian mixture $p(x_1, x_2) = \delta(x_2) \sum_{i=1}^{5} \frac{1}{5} (2\pi v_i)^{-\frac{1}{2}} \exp(-\frac{(x_1 - \mu_i)^2}{2v_i})$ with equal weights, variances $v = (0.2, 0.5, 1, 2, 4)$, and means $\mu = (-20, -10, 0, 10, 20)$. Extended score is applied with parameters $a = b = p = 1$.

In Figure 2a we consider a mixture of Gaussians with covariance matrices $\Sigma_i = R_i \text{diag}(1.7, 0.2)$, where the $R_i$, $i \in \{1, \ldots, 21\}$, are randomly chosen rotation matrices, i.e. around each mean of the mixture we have large variance in one direction and small variance in the direction orthogonal to the first one. Locally this can be viewed as a $1.5$-variance Gaussian along some $1$-dimensional affine linear subspace to which Gaussian noise with covariance matrix $0.2\mathbb{I}$ is added, resulting in a $1.7$ variance in one direction and a $0.2$ variance in the other. The denoising effect is clear: the extended score inference moves the points onto the affine linear subspace corresponding to the first principal direction of each Gaussian, while leaving the spread along the affine linear subspace almost unchanged. In Figure 2b we demonstrate that this effect is not limited to affine linear manifolds, by considering a mixture of radial Gaussians of the form $p(x) = C \sum_{i=1}^{5} \exp(-\frac{(\|x - \mu_i\| - r)^2}{2v})$ with variance $v = 2.5$, radius $r = 10$, randomly chosen means $\mu_i$, and $C$ a normalizing constant.

## 5.2 Effects on FFHQ, AFHQv2, and ImageNet

While the last section provides some basic intuition for the effects of our inference method, it is, of course, not so clear how this translates to much more complicated distributions in much higher dimensions. We will explore this question using our inference method with the pretrained score networks from Karras et al. (2022) for[3] FFHQ (Karras et al. (2019)), AFHQv2 (Choi et al. (2020)), and ImageNet (Deng et al. (2009)), where we also use the time schedule suggested in Karras et al. (2022), and only vary the hyperparameters specific to our method, i.e. $a$, $b$, $p$, and $\delta$ in Algorithm 1. Since these datasets are arguably essentially noiseless, the effect of the extended score will be visible on the different features of the images. The following two sections consider noisy datasets.

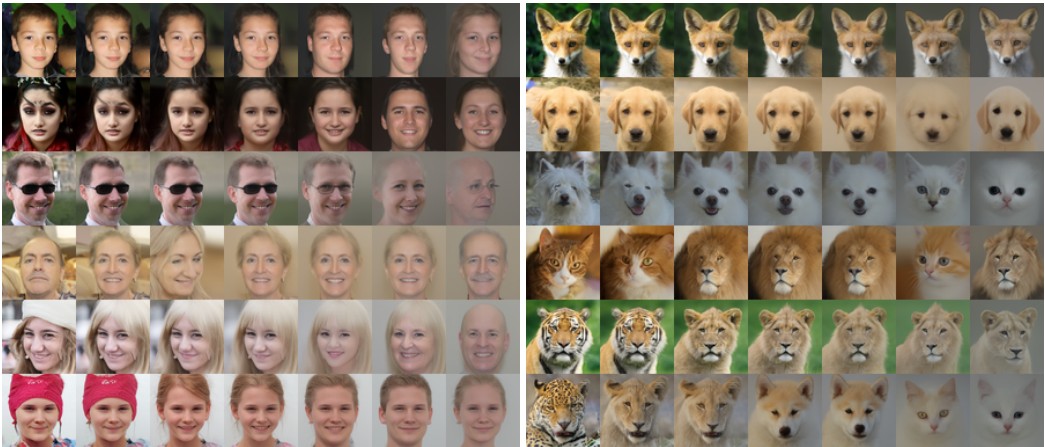

Figure 3: In both subfigures, all images in each row start from the same latent noise sample, and the leftmost column uses standard score whereas second to last columns use Algorithm 1 with $\delta = 0.0001$, $p = 8$, $a = 2.5$, and, left to right, $b \in \{2, 5, 10, 20, 40, 80\}$. Further examples in Figure 10.

In Figure 3, we showcase the impact of our inference methods on the generation of human or animal faces using the pretrained score networks for FFHQ and AFHQv2, respectively. Note that increasing the parameter $b$ in Algorithm 1 results in a greater impact of the extended score, i.e. we expect stronger attraction to the manifold of primary variation. We observe that for all values of $b$, we generate samples with qualitatively the same facial features, but for larger values of $b$, we always get a plain single-colored background. In case of faces, it seems quite clear that the direction of primary relevance should correspond to essential facial features that are present in all the data, whereas the background variation, as well as features like glasses and head wear, are split across a much larger number of directions and are thresholded out first. We also observe similar effects on ImageNet, e.g. in the examples in Figure 4, where the extended score inference seems to focus on generating one primary object, while progressively thresholding out everything else as we increase $b$. However, since ImageNet contains a large variety of objects, the primary directions differ between classes, and thus the effect of extended score inference with a given choice of hyperparameters is much more varied across different starting noise images (see Figure 11 in the Appendix).

Although Algorithm 1 is deterministic when started with the same random seed, i.e. such that it starts from the same latent noise image, it relies on a score network trained through a highly stochastic process. Consequently, even small changes of the iterates $x_i$ can build up and lead to the generation of a significantly different image. As can be seen in Figure 3, in general the algorithm is not particularly suited for orthogonal projection on the manifold of, in this case, faces, i.e. it does not simply generate the same face as standard inference would but without background. However, as showcased in Figure 9 in the Appendix, there are often certain ranges of parameters that lead to rather similar images and may be used for manual adjustment of a generated image.

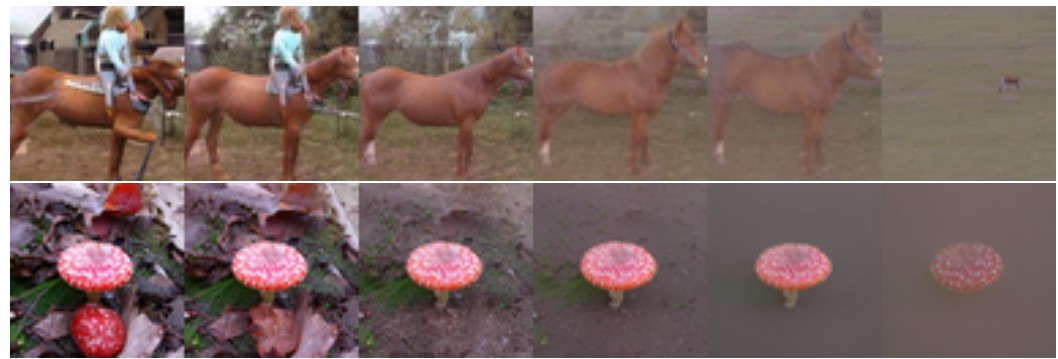

Figure 4: Each row starts from the same latent noise sample, and the leftmost column uses standard score whereas second to last columns use Algorithm 1 with $\delta = 0.001$, $p = 12$, and $a = 8$, $b \in \{5, 20, 50, 80, 250\}$ for the top row, $a = 4$, $b \in \{2, 10, 30, 60, 100\}$ for the bottom row.

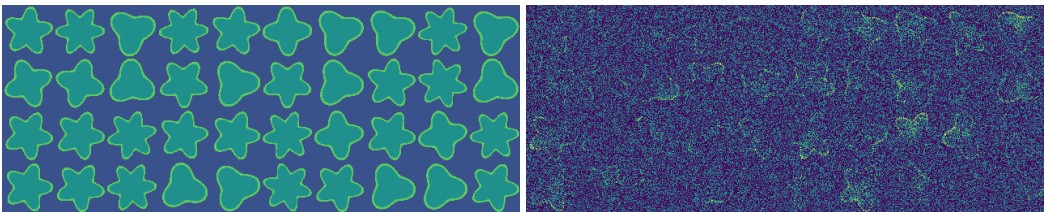

Figure 5: Grid of clean samples from the synthetic data set (left); grid of corresponding noisy samples (right). Each sample is an 8-bit grayscale $64 \times 64$ image, displayed via viridis colormap.

### 5.3 DENOISING SYNTHETIC DATA

We created a synthetic dataset of clean samples, added noise, and trained a diffusion model only on the noisy data. Our clean distribution consists of 8-bit grayscale $64 \times 64$ images depicting 4 different shapes, each appearing with equal probability and rotated by an angle chosen uniformly at random (see Figure 5, left). As such, this probability distribution is essentially supported on 4 disjoint 1-dimensional manifolds in pixel space. We then add two types of corruption to obtain noisy samples. Firstly, we blot out large parts of the shape, by uniformly randomly picking 50 locations on the boundary of the shape and subtracting Gaussian bump functions centered at those locations from the clean image. Secondly, we add i.i.d. Gaussian noise to each pixel, resulting in images where the original shapes can hardly be discerned from a single image (see Figure 5, right). We then trained a DDPM++ model[4] on a set of $100\,000$ of such noisy samples for a duration of 3 mimg, i.e. 30 repeats per image. When using the trained network for standard score inference we obtain a reproduction of the corrupted samples (see Figure 6, right). In contrast, running our extended score inference, we obtain samples showing the original shapes (see Figure 6, left). Figure 12 in the Appendix illustrates the dependence on the extended score parameters $\delta$, $a$, $b$, and $p$. In order to test whether our method works on corrupted training data beyond the case of Gaussian noise, we also trained a DDPM++ model, using the same training settings, on a set of $100\,000$ synthetic samples, where each pixel was set to 0 with probability 0.5 (see Figure 14 in Appendix A.3). As can be seen in Table 1, for both cases extended score inference leads to a significant improvement in FID compared to the standard score, which generates samples with essentially the same distance to the clean data as the noisy data.

### 5.4 DENOISING CIFAR-10

Next, we test whether the denoising capabilities of our method extend to real data. Specifically we train a DDPM++ model on CIFAR-10 with additive Gaussian noise. While standard score inference

---

[3]Note that these networks have been trained on images from these datasets which have been downsampled to a $64 \times 64$ resolution.

[4]Using the method from the accompanying github to Karras et al. (2022) with default settings except for cond $= 0$ and augment $= 0$, which took roughly 14 hours on a single $A100$ GPU.

| Noisy dataset | training data | standard generation | extended generation |
|---|---|---|---|
| CIFAR-10, Gaussian noise | 136.87 | 147.47 | **77.15** |
| synthetic data, Gaussian noise | 320.56 | 319.42 | **189.51** |
| synthetic data, pixel removal | 233.86 | 234.40 | **163.14** |

Table 1: FIDs with respect to the corresponding clean dataset, computed from $10\,000$ images.

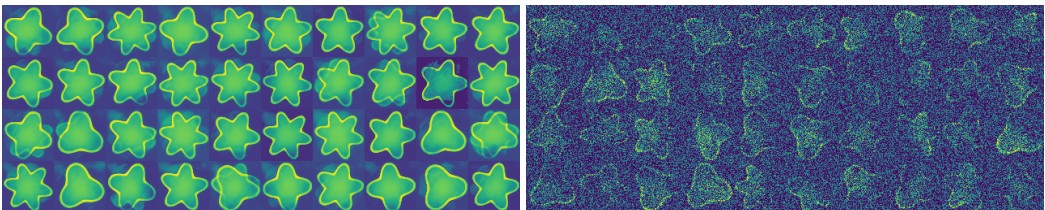

Figure 6: Grid of samples generated from consecutive random seeds by Algorithm 1 with parameters $\delta = 0.02$, $p = 8$, $a = 0.002$, and $b = 15$ (left); grid of samples generated by inference with standard score (right). Each sample is an 8-bit grayscale $64 \times 64$ image, displayed via viridis colormap.

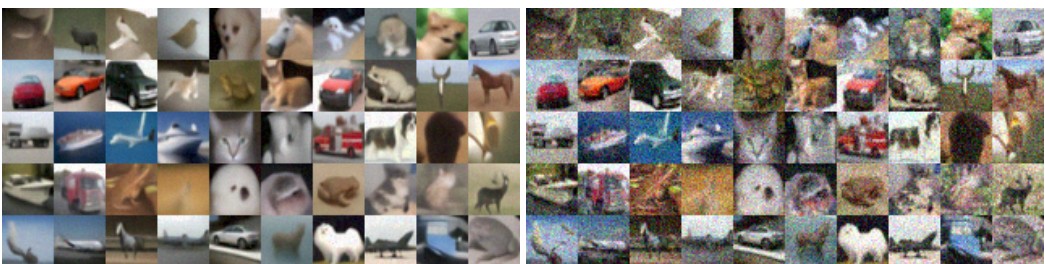

Figure 7: Trained on CIFAR-10 with additive Gaussian noise and generated by standard score inference (right) or by extended score inference (left).

produces image with roughly the amount of noise of the images it was trained on, extended score inference produces significantly cleaner images, both visually (Figure 7) as well as in the FID scores (Table 1). The intermediate images generated during inference (see Figure 15 in Appendix A.3) strongly indicate that this effect cannot simply be achieved by truncated sampling.

## 5.5 DENOISING REAL DATA

In this section we test our method on real data from single-particle Cryo-Electron Microscopy (see Cheng et al. (2015) for an introduction), where many particles of the same type are suspended in liquid, frozen, and put under an electron microscope. This produces extremely noisy 2D-images which need to be refined before further steps like 3D reconstruction can be attempted. This presents an opportunity to investigate the performance of our method on practically relevant real world data with non-Gaussian noise. We use the EMPIAR-11618 (Bacic et al., 2021) dataset of $68\,401$ grayscale images with $256 \times 256$ resolution. They were extracted from raw data and undergone some preprocessing, but are still very noisy (see Figure 8, top left). We trained a DDPM++ model[5] on this data and used our method to generate samples (see Figure 8, right) whose shapes correspond strongly to what has been obtained by Bacic et al. (2021), see Figure 8, bottom. We emphasize that the network has only ever seen noisy data and has in no way been specifically adjusted based on a priori knowledge of these shapes. This can be seen by the fact that standard score inference generates noisy samples, very similar to those in the training set (see Figure 16 in the Appendix). The parameters used for Figure 8 of the extended score inference have been determined by hand with such knowledge, of course, but similar results are generated for a wide range of parameter choices (see

---

[5]In order to compensate for the higher resolution we reduced the number of feature channels in the ddpmpp architecture from $128$ to $32$, but otherwise used the same settings as for the synthetic data. Training took roughly 60 hours on two $A100$ GPUs.

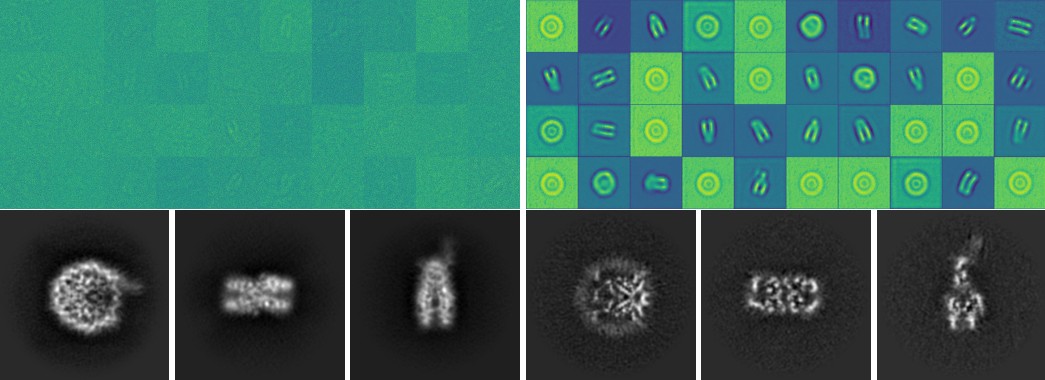

Figure 8: Images from the EMPIAR-11618 dataset (left). Samples generated by Algorithm 1 with parameters $\delta = 0.001$, $p = 8$, $a = 0.01$, and $b = 2$ (right). Shapes obtained by Bacic et al. (2021) (bottom), taken from (wwPDB Consortium, 2023, EMD-17944). The colormap is applied with normalization per image to enhance contrast.

Figure 13 in the Appendix). In particular, this demonstrates a significant capability of the extended score to guide generation towards samples from the underlying image manifold, also in the case of real data with extreme noise corruption caused by physical measurement modalities. This example serves as proof of concept for the validation of our approach: reaching state-of-the-art performance is outside the scope of this work, and would require incorporating more prior domain knowledge.

## 6 CONCLUSION

We introduced Manifold Attracted Diffusion (MAD), a novel inference approach for score-based diffusion models to generate clean samples from a distribution despite training on noisy datasets. Based on the manifold hypothesis, our method utilizes the underlying geometry to suppress off-manifold variations while preserving on-manifold variations, which results in attracting samples toward a low-dimensional structure. It can leverage established training algorithms and pretrained models. The required computation cost is at most twice that of standard inference, due to requiring a second evaluation of the score network. However, we observed empirically that it is enough to use the extended score only in the last half of the inference steps, yielding an additional computational cost of 50%, but additional investigation on this aspect is required. Numerical experiments on both synthetic and real data demonstrate that MAD successfully suppresses noise.

Future directions include extending MAD to solve inverse problems, as in Cryo-EM for denoising individual images, or for other denoising or image restoration tasks. This would benefit from integration with conditional diffusion models, as explored in diffusion posterior sampling frameworks (Chung et al., 2023; 2022). Developing automatic and adaptive parameter selection for $\gamma(t)$ would enhance the methods robustness and applicability. Further theoretical analysis of the extended score may lead to an improved incorporation into the inference procedure, in particular one could combine it with inference techniques using noise injection or higher order ODE discretization schemes. Future work should also involve a rigorous quantitative study comparing the performance and computational trade-offs of MAD against training-time methods, finetuning on limited clean data, and other relevant inference-time baselines such as truncated sampling (Daras et al., 2025). Furthermore, it would be valuable to investigate whether the MAD framework could be adapted to time-unconditioned generative models (Sun et al., 2025), or if our method fundamentally relies on a time-dependent score. Finally, a promising future direction, motivated by the conceptual similarities to classifier-free guidance (Ho & Salimans, 2021), is to explore whether our extended score can serve as a more interpretable, geometrically-grounded method for improving general generation quality, even for models trained on clean data.

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

# A APPENDIX

## A.1 THE EXTENDED SCORE FOR PRODUCTS OF MEASURES

In the following lemma, we derive the expression of the extended score of the product of two measures. In this section, the superscripts denote the dimension of the domain.

**Lemma A.1.** *Let $d_1 + d_2 = d$. Take $p_1 \in \widetilde{M}(\mathbb{R}^{d_1})$ and $p_2 \in \widetilde{M}(\mathbb{R}^{d_2})$. Then the product measure $p = p_1 \otimes p_2$ belongs to $\widetilde{M}(\mathbb{R}^d)$ and*

$$H_0^d p(x) = (H_0^{d_1} p_1(x_1), H_0^{d_2} p_2(x_2)),$$

*where $x = (x_1, x_2) \in \mathbb{R}^{d_1 + d_2}$.*

*Proof.* We have that $g_\gamma^d(x) = g_\gamma^{d_1}(x_1) g_\gamma^{d_2}(x_2)$. Thus, by Fubini's theorem, we have

$$(p * g_\gamma^d)(x) = (p_1 * g_\gamma^{d_1})(x_1) \cdot (p_2 * g_\gamma^{d_2})(x_2).$$

Thus

$$\log(p * g_\gamma^d)(x) = \log(p_1 * g_\gamma^{d_1})(x_1) + \log(p_2 * g_\gamma^{d_2})(x_2).$$

Taking a gradient with respect to $x$ we obtain

$$S^d(p * g_\gamma^d)(x) = (S^{d_1}(p_1 * g_\gamma^{d_1})(x_1), 0_{d_2}) + (0_{d_1}, S^{d_2}(p_2 * g_\gamma^{d_2})(x_2)).$$

Therefore, since the expression of $H_\gamma^d(p)$ is linear in $S^d(p * g_\gamma^d)$, we obtain

$$H_\gamma^d(p)(x) = (H_\gamma^{d_1}(p_1)(x_1), 0_{d_2}) + (0_{d_1}, H_\gamma^{d_2}(p_2)(x_2)) = (H_\gamma^{d_1}(p_1)(x_1), H_\gamma^{d_2}(p_2)(x_2)).$$

Taking the limit as $\gamma \to 0$, the result follows. $\square$

This result can be used to calculate the extended score of a degenerate distribution $p$, namely, a distribution supported on a lower-dimensional affine subspace of $\mathbb{R}^d$. Since the score, and thus the extended score, is equivariant with respect to rotations and translations, without loss of generality, we can assume that $p$ is supported on $\{(x_1, 0_{d_2}) \in \mathbb{R}^d : x_1 \in \mathbb{R}^{d_1}\}$ with density $p_1 \in P(\mathbb{R}^{d_1})$, namely,

$$p = p_1 \otimes \delta_{d_2},$$

where $\delta_{d_2} \in \widetilde{M}(\mathbb{R}^{d_2})$ is the Dirac delta centered at 0 in $\mathbb{R}^{d_2}$. By Lemmata 3.2 and A.1, we obtain that the extended score of $p$ is given by

$$H_0^d p(x) = (H_0^{d_1} p_1(x_1), H_0^{d_2} p_2(x_2)) = (S^{d_1} p_1(x_1), -x_2),$$

where we have also used that $H_0\delta(x) = -x$.

In the particular case when $p_1$ is a non-degenerate Gaussian distribution on $\mathbb{R}^{d_1}$ with mean $\mu_1 \in \mathbb{R}^{d_1}$ and covariance $\Sigma_1 \in \mathbb{R}^{d_1 \times d_1}$, namely $p_1 = \mathcal{N}(\mu_1, \Sigma_1)$, we have

$$H_0^d p(x) = (-\Sigma_1^{-1}(x_1 - \mu_1), -x_2).$$

As expected, this coincides with the standard score of the (non-degenerate) Gaussian distribution $\mathcal{N}((\mu_1, 0_{d_2}), \Sigma)$ on $\mathbb{R}^d$, where $\Sigma$ is the block matrix given by

$$\Sigma = \begin{bmatrix} \Sigma_1 & 0 \\ 0 & \mathbb{I}^{d_2} \end{bmatrix}.$$

### A.2 PROOFS AND DERIVATIONS

*Proof of Lemma 3.2.* By standard properties of approximate identities (Grafakos, 2014, Example 1.2.17 and Theorem 1.2.19(2)), for $f \in C(\mathbb{R}^d)$ bounded we have that

$$\lim_{\gamma \to 0} (f * g_\gamma)(x) = f(x), \qquad x \in \mathbb{R}^d,$$

Analogously, with $h_\gamma(x) := \frac{1}{\gamma}\|x\|^2 g_\gamma(x)$ by (Grafakos, 2014, Theorem 1.2.21(b)) we get that

$$\lim_{\gamma \to 0} (f * h_\gamma)(x) = C f(x), \qquad x \in \mathbb{R}^d,$$

where $C = \int_{\mathbb{R}^d} (2\pi)^{-\frac{d}{2}} \|z\|^2 e^{-\frac{\|z\|^2}{2}} \, dz < \infty$. In addition, we observe that

$$\frac{\partial}{\partial x_i} g_\gamma(x) = -\frac{x_i}{\gamma} g_\gamma(x)$$

and

$$\frac{d}{d\gamma} g_\gamma(x) = \left(\frac{-d}{2\gamma} + \frac{\|x\|^2}{2\gamma^2}\right) g_\gamma(x) = \frac{1}{2\gamma}(h_\gamma(x) - d g_\gamma(x)).$$

Consequently, as $p(x) > 0$ for all $x \in \mathbb{R}^d$, by assumption

$$
\begin{aligned}
\lim_{\gamma \to 0} \gamma \frac{d}{d\gamma} S(p * g_\gamma) &= \lim_{\gamma \to 0} \gamma \frac{d}{d\gamma} \frac{\nabla_x(p * g_\gamma)}{p * g_\gamma} = \lim_{\gamma \to 0} \gamma \frac{d}{d\gamma} \frac{\nabla_x p * g_\gamma}{p * g_\gamma} \\
&= \lim_{\gamma \to 0} \gamma \frac{(p * g_\gamma)(\nabla_x p * \frac{d}{d\gamma} g_\gamma) - (\nabla_x p * g_\gamma)(p * \frac{d}{d\gamma} g_\gamma)}{(p * g_\gamma)^2} \\
&= \lim_{\gamma \to 0} \frac{(p * g_\gamma)\big((\nabla_x p * h_\gamma) - d(\nabla_x p * g_\gamma)\big) - (\nabla_x p * g_\gamma)\big((p * h_\gamma) - d(p * g_\gamma)\big)}{2(p * g_\gamma)^2} \\
&= \frac{p(C\nabla_x p - d\nabla_x p) - \nabla_x p(Cp - dp)}{2p^2} \\
&= 0.
\end{aligned}
$$

Moreover, it holds that

$$\lim_{\gamma \to 0} S(p * g_\gamma) = \lim_{\gamma \to 0} \frac{\nabla_x(p * g_\gamma)}{p * g_\gamma} = \frac{\lim_{\gamma \to 0} \nabla_x p * g_\gamma}{\lim_{\gamma \to 0} p * g_\gamma} = \frac{\nabla_x p}{p} = Sp,$$

which completes the proof. $\qquad\square$

*Proof of Lemma 3.3.* We write $h_i(x) := c_i e^{-\frac{\|x - \mu_i\|^2}{2\gamma}}$, i.e. $p * g_\gamma = \sum_{i \in [n]} (2\pi\gamma)^{-\frac{d}{2}} h_i$, and observe that

$$S(p * g_\gamma)(x) = -\sum_{i \in [n]} \frac{x - \mu_i}{\gamma} w_i(x),$$

where

$$w_i := \frac{h_i}{\sum_{j \in [n]} h_j}.$$

We observe that

$$\frac{\mathrm{d}}{\mathrm{d}\gamma}w_i(x) = \frac{(\sum_{j\in[n]}h_j(x))\frac{\|x-\mu_i\|^2}{2\gamma^2}h_i(x) - h_i(x)\sum_{j\in[n]}\frac{\|x-\mu_j\|^2}{2\gamma^2}h_j(x)}{(\sum_{j\in[n]}h_j(x))^2}$$

and

$$H_\gamma p(x) = (1+\gamma)S(p*g_\gamma)(x) + \gamma\frac{\mathrm{d}}{\mathrm{d}\gamma}S(p*g_\gamma)(x) = \gamma S(p*g_\gamma)(x) + \frac{\mathrm{d}}{\mathrm{d}\gamma}\gamma S(p*g_\gamma)(x).$$

We will first show that the second term vanishes for $\gamma \to 0$. To this end, we note

$$\frac{\mathrm{d}}{\mathrm{d}\gamma}\gamma S(p*g_\gamma)(x) = -\sum_{i\in[n]}(x-\mu_i)\frac{(\sum_{j\in[n]}h_j(x))\frac{\|x-\mu_i\|^2}{2\gamma^2}h_i(x) - h_i(x)\sum_{j\in[n]}\frac{\|x-\mu_j\|^2}{2\gamma^2}h_j(x)}{(\sum_{j\in[n]}h_j(x))^2}$$

$$= -\frac{\sum_{i,j\in[n]}h_i(x)h_j(x)(x-\mu_i)(\|x-\mu_i\|^2 - \|x-\mu_j\|^2)}{2\gamma^2(\sum_{j\in[n]}h_j(x))^2}.$$

Let $x \in W_k$ and $i,j \in [n]$ such that $x \notin W_i \vee x \notin W_j$, then

$$\lim_{\gamma\to 0}\frac{h_i(x)h_j(x)}{\gamma^2(\sum_{j\in[n]}h_j(x))^2} \le \lim_{\gamma\to 0}\frac{h_i(x)h_j(x)}{\gamma^2 h_k(x)^2} = \lim_{\gamma\to 0}\frac{c_i c_j}{\gamma^2 c_k^2}e^{\frac{2\|x-\mu_k\|^2 - \|x-\mu_i\|^2 - \|x-\mu_j\|^2}{2\gamma}} = 0$$

as $2\|x-\mu_k\|^2 - \|x-\mu_i\|^2 - \|x-\mu_j\|^2 < 0$ by definition of $W_i$. Since $\|x-\mu_i\|^2 - \|x-\mu_j\|^2 = 0$ if[6] $x \in W_i \cap W_j$, and $h_i(x) \ge 0$ for every $x \in \mathbb{R}^d$, $i \in [n]$, we have

$$\lim_{\gamma\to 0}\frac{\mathrm{d}}{\mathrm{d}\gamma}\gamma S(p*g_\gamma)(x) = 0.$$

We proceed by noting that

$$\lim_{\gamma\to 0}\frac{h_j(x)}{h_i(x)} = \lim_{\gamma\to 0}\frac{c_j}{c_i}e^{-\frac{\|x-\mu_j\|^2 - \|x-\mu_i\|^2}{2\gamma}} = \begin{cases} 0, & \|x-\mu_i\| < \|x-\mu_j\| \\ \frac{c_j}{c_i} & \|x-\mu_i\| = \|x-\mu_j\| \\ \infty, & \|x-\mu_i\| > \|x-\mu_j\| \end{cases}$$

and, consequently, using the conventions that $\frac{1}{0} = \infty$ and $\frac{1}{\infty} = 0$,

$$\lim_{\gamma\to 0}w_i(x) = \lim_{\gamma\to 0}\frac{h_i(x)}{\sum_{j\in[n]}h_j(x)}$$

$$= \lim_{\gamma\to 0}\left(1 + \sum_{j\in[n],j\neq i}\frac{h_j(x)}{h_i(x)}\right)^{-1}$$

$$= \begin{cases} 0, & x \notin W_i \\ 1, & (x \in W_i) \wedge (x \notin W_j \forall j \in [n]\setminus\{i\}) \\ (\sum_{j\in J}\frac{c_j}{c_i})^{-1}, & i \in J \subseteq [n]\colon x \in \bigcap_{j\in J}W_j(x) \end{cases}.$$

Thus

$$H_0 p(x) = \lim_{\gamma\to 0}H_\gamma p(x) = \lim_{\gamma\to 0}\gamma S(p*g_\gamma)(x) = -\sum_{i\in[n]}(x-\mu_i)z_i(x).$$

This concludes the proof. □

*Derivation of equation (7).* For $S_\theta(\sigma,x) = -\frac{x-\mu}{\sigma^2}$, we have

$$\frac{\mathrm{d}}{\mathrm{d}\sigma}S_\theta(\sigma,x) = \frac{2(x-\mu)}{\sigma^3}$$

---

[6]In particular, if $i = j$.

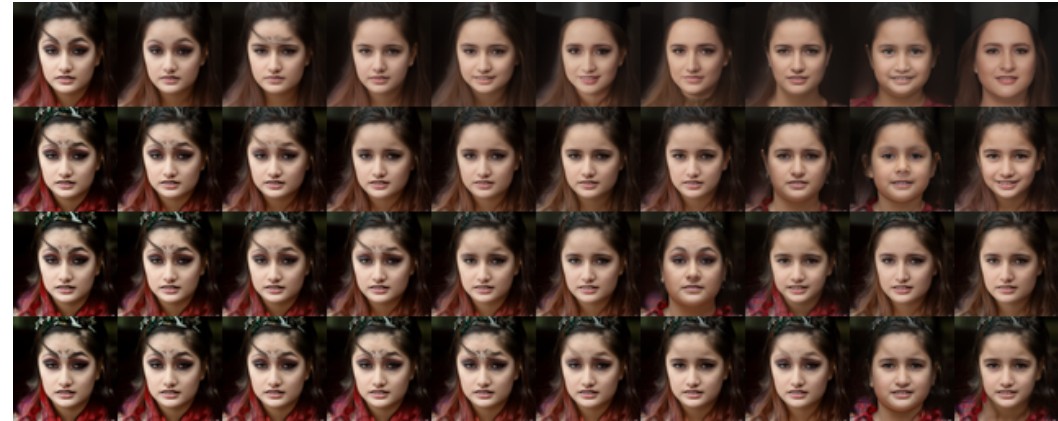

Figure 9: Generated with Algorithm 1 from the same latent noise sample with $\delta = 0.0001$, $p = 8$, and, from left to right, $b \in \{1, 2, 3, 4, 5, 6, 7, 8, 9, 10\}$ as well as, from top to bottom, $a \in \{1, 3, 5, 7\}$.

and thus (5) evaluates to

$$x_{i+1} = x_i - m(t_i)(t_{i+1} - t_i)t_i\big((1 + \gamma(t_i))(-\tfrac{x_i - \mu}{t_i^2}) + \tfrac{b\gamma(t_i)}{2t_i}\tfrac{2(x_i - \mu)}{t_i^3}\big)$$
$$= x_i - m(t_i)\tfrac{t_i - t_{i+1}}{t_i}(x_i - \mu)\big(1 + \gamma(t_i) - \tfrac{b\gamma(t_i)}{t_i^2}\big).$$

We would like to force this to match the standard score inference step in (6), for this special case of $S_\theta(\sigma, x)$, which is achieved by choosing

$$m(t_i) = (1 + \gamma(t_i) - \tfrac{b\gamma(t_i)}{t_i^2})^{-1}.$$

$\square$

## A.3 ADDITIONAL NUMERICAL EXAMPLES

Additional examples related to the datasets FFHQ and AFHQv2 are shown in Figure 10. Additional examples related to ImageNet are shown in Figure 11.

Figure 12 and Figure 13 show that the samples generated by extended score inference have a strong dependence on the chosen hyperparameters, but a significant emergence of structure can be observed for many different choices. Despite being a simpler problem, the generation seems to be less stable w.r.t. hyperparameter choices for the synthetic data. One reason for this might be that we have a 1-dimensional manifold of images that is harder to find than the one underlying the EMPIAR-11618 data, which is, of course, not explicitly known but can be assumed to be higher-dimensional. It may also be due to a shorter training duration (3mimg compared to 10mimg), which would be consistent with the generation being much more stable w.r.t. the hyperparameters for FFHQ, AFHQv2, and ImageNet as the networks for those problems have been trained significantly longer.

In Figure 14, we show the results with the synthetic dataset with 50% pixel removal. We see that images generated by standard score inference replicate the corruption present in the training set, while the images generated by extended score inference exhibit all key features of the clean data, while eliminating the corruption due to pixel removal.

In Figure 15, we compare the inference paths for standard and extended score generations: applying truncated sampling to the standard score generation would not be enough to obtain the denoising effect.

Additional images from the EMPIAR-11618 dataset, as well as images generated by standard score inference, are shown in Figure 16.

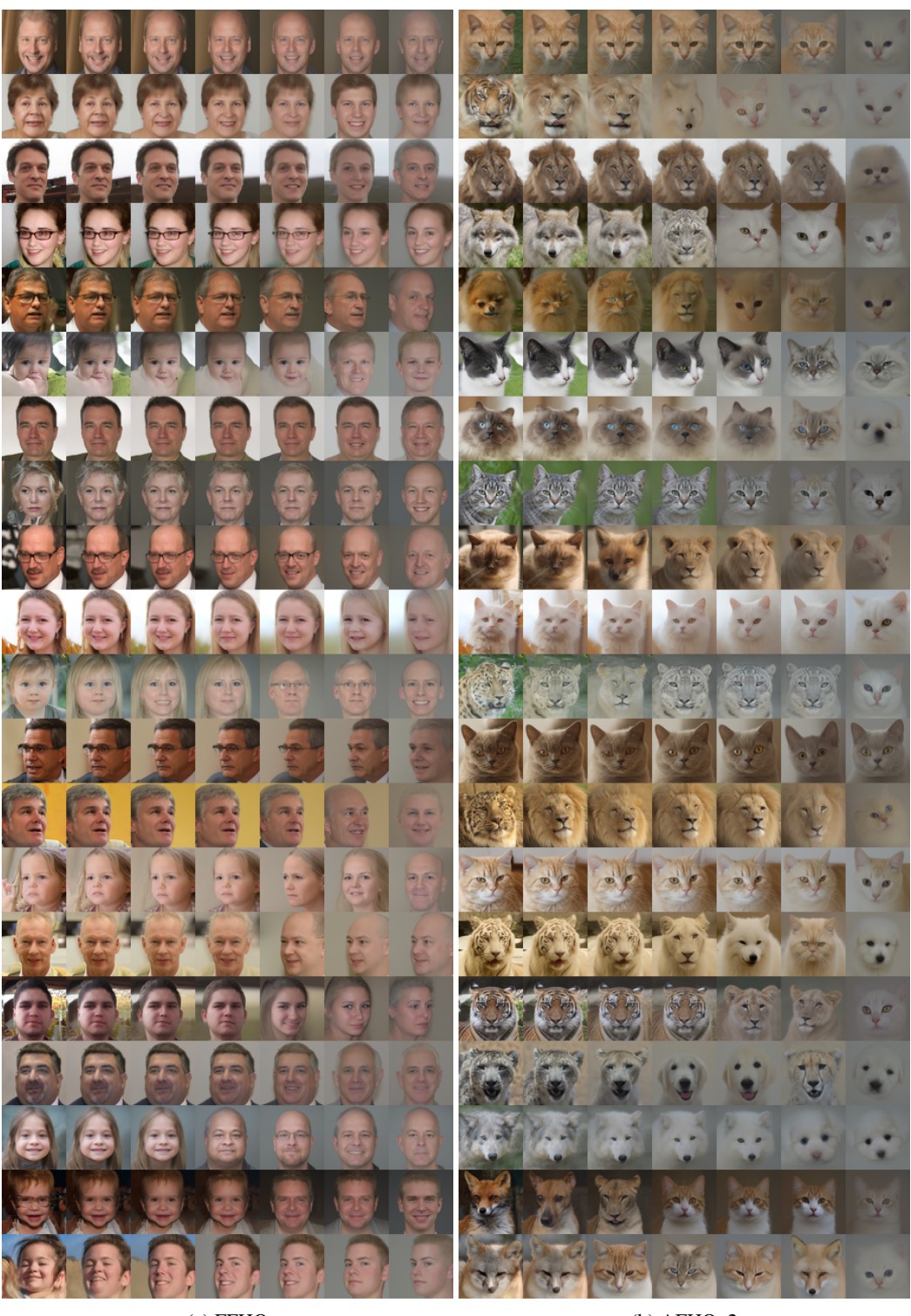

(a) FFHQ                    (b) AFHQv2

Figure 10: In both subfigures, all images in each row starts from the same latent noise sample and the leftmost column uses standard score whereas second to last columns use Algorithm 1 with $\delta = 0.0001$, $p = 8$, $a = 2.5$, and, from left to right, $b \in \{2, 5, 10, 20, 40, 80\}$. The rows are generated from consecutive random seeds.

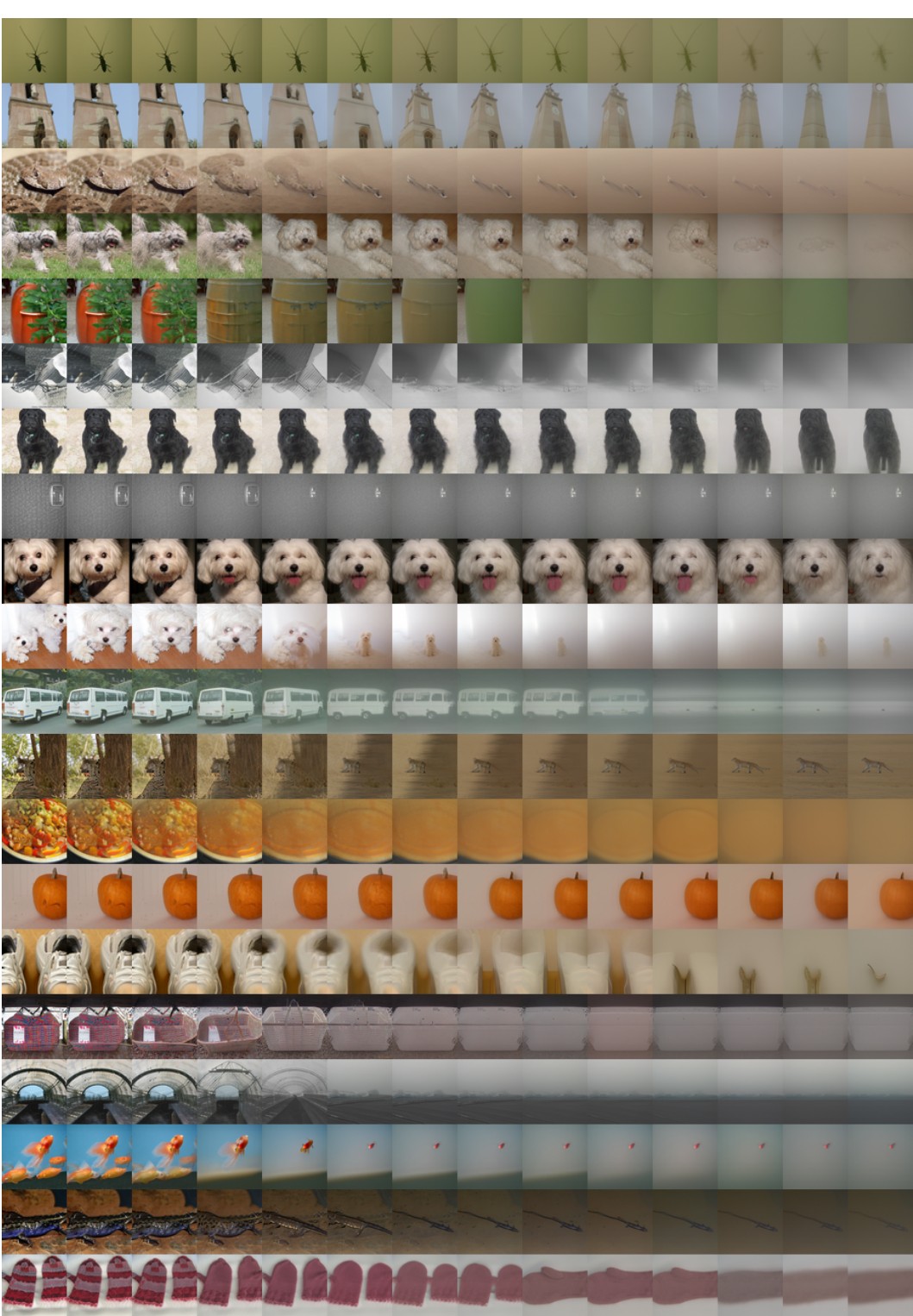

Figure 11: All images in each row starts from the same latent noise sample and the leftmost column uses standard score whereas second to last columns use Algorithm 1 with $\delta = 0.001$, $p = 12$, $a = 4$, and, from left to right, $b \in \{1, 2, 5, 10, 15, 20, 25, 30, 35, 40, 50, 60, 70\}$. The rows are generated from consecutive random seeds.

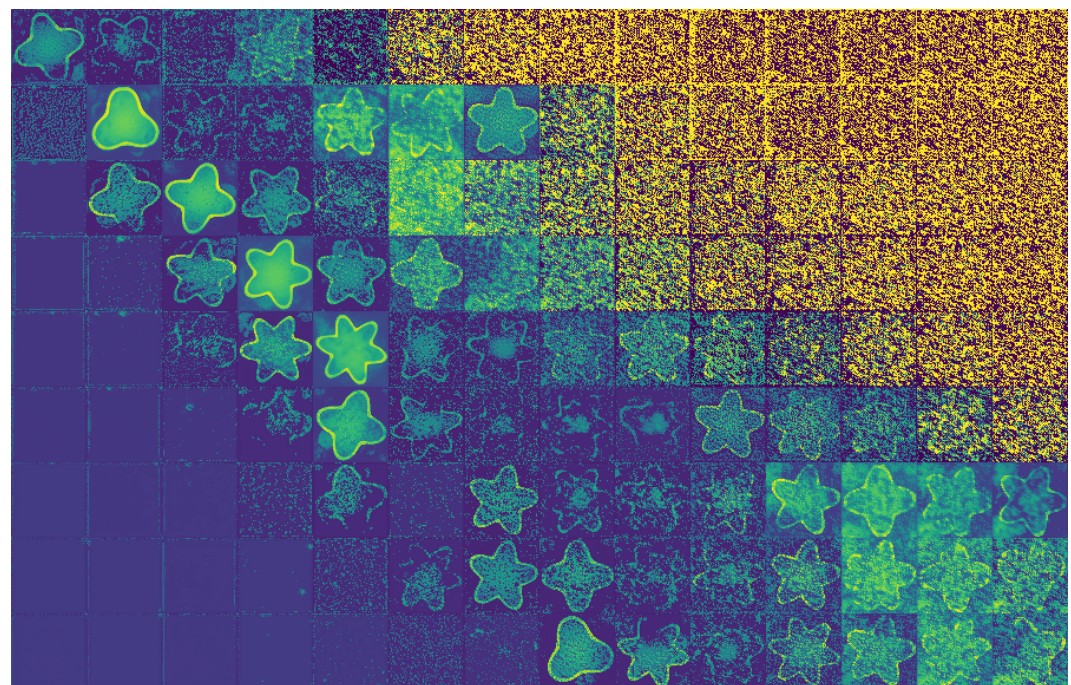

Figure 12: Generated with Algorithm 1 from the same latent noise sample with $\delta = 0.02$, $p = 8$, and, from left to right, $b \in \{10, 15, 20, 25, 30, 35, 40, 45, 50, 55, 60, 65, 70, 75\}$ as well as, from top to bottom, $a \in \{0.0015, 0.002, 0.0025, 0.003, 0.0035, 0.004, 0.0045, 0.005, 0.0055\}$.

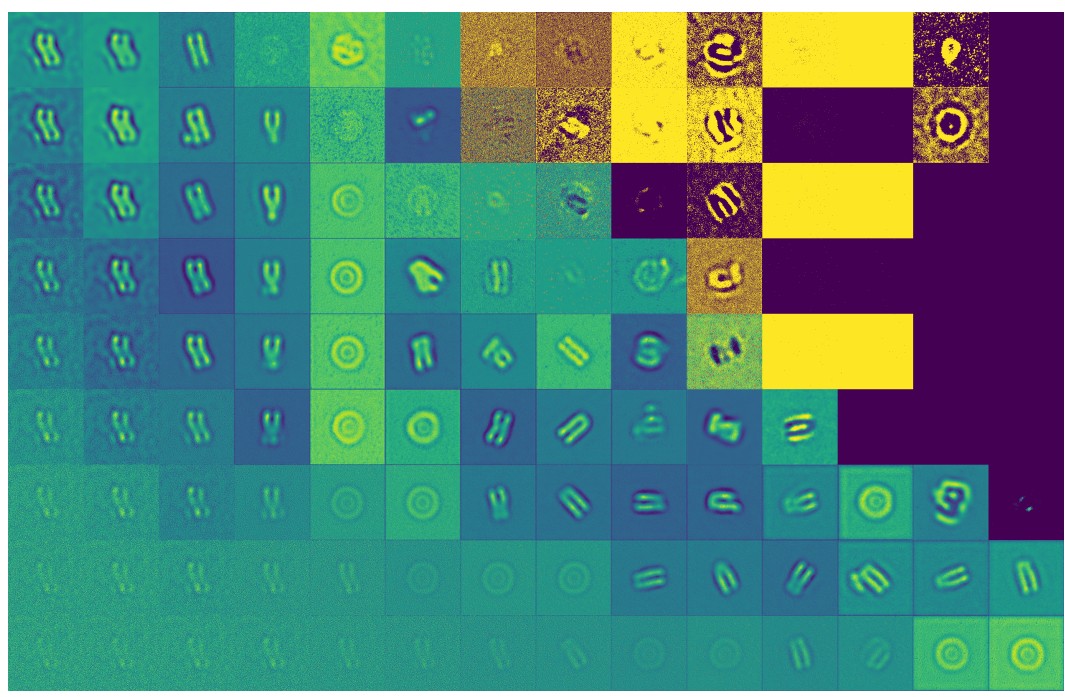

Figure 13: Generated with Algorithm 1 from the same latent noise sample with $\delta = 0.001$, $p = 8$, and, from left to right, $b \in \{0.25, 0.5, 1, 2, 4, 8, 16, 32, 64, 128, 256, 512, 1024, 2048\}$ as well as, from top to bottom, $a \in \{0.0005, 0.001, 0.002, 0.005, 0.01, 0.02, 0.05, 0.1, 0.2\}$.

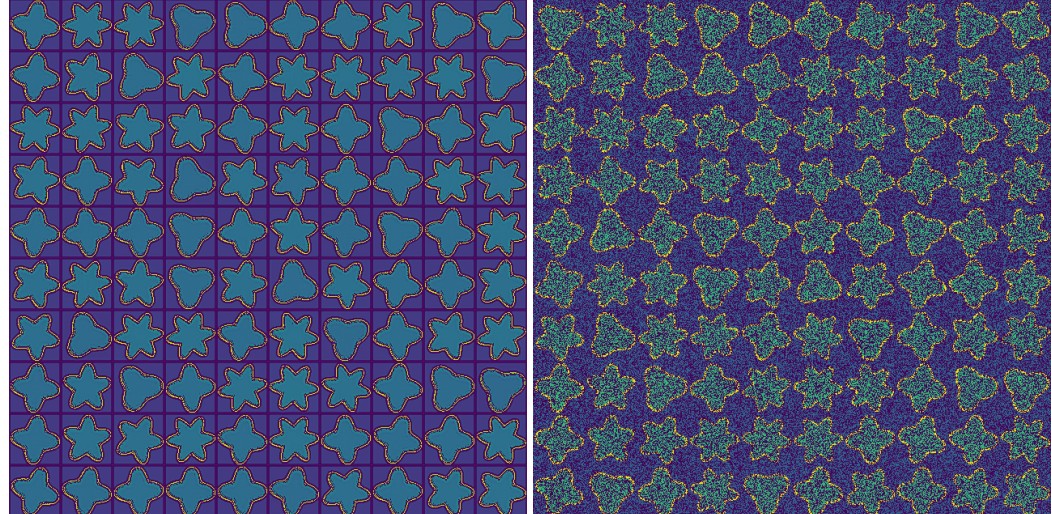

Figure 14: Comparison of generated images using a network trained on our synthetic dataset with pixel removal noise, specifically in a given image every pixel is set to $0$ with probability $0.5$. Images generated by standard score inference (right). Images generated by extended score inference (left).

## A.4 LLM USE

LLMs were used to polish the writing for parts of the text, to suggest related work, and as a coding aid. All those suggestions have only been implemented after thorough manual review. No LLMs were involved in any way in the mathematical derivations.

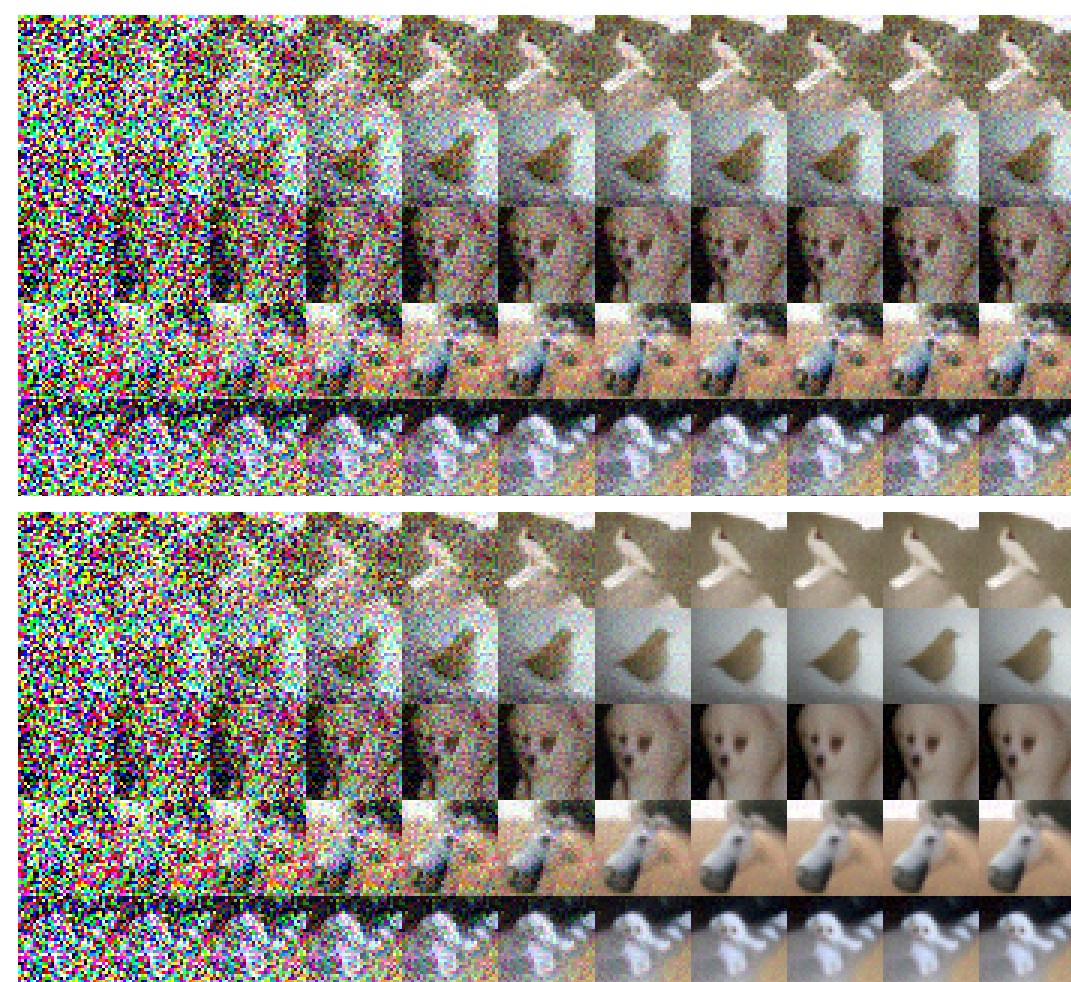

Figure 15: Illustrating the inference path for standard score generation (top) compared to extended score generation (bottom), displaying every second image generated during the inference procedure starting at step 20.

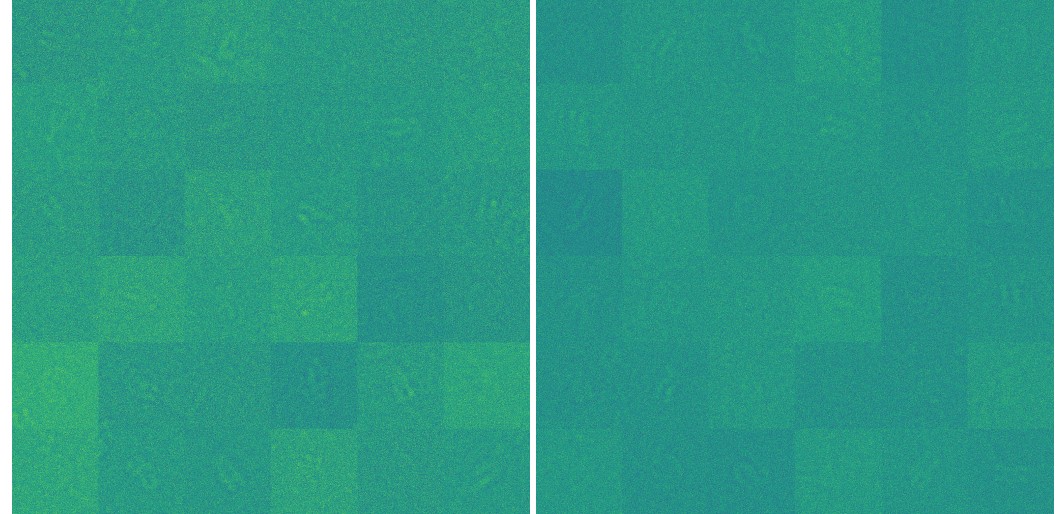

Figure 16: Images from the EMPIAR-11618 dataset (left). Images generated by standard score inference (right).

