# OpenReview forum: "MAD: Manifold Attracted Diffusion"
_ICLR.cc/2026/Conference — Submitted to ICLR 2026_

### Official Review · Reviewer_4Etn · 2025-10-31

**Soundness:** 2
**Presentation:** 3
**Contribution:** 2
**Rating:** 4
**Confidence:** 3

**Summary:**

This paper presents MAD (Manifold Attracted Diffusion) which is designed for denoising and structure recovery when diffusion models are trained on corrupted or noisy datasets. It modifies the inference dynamics to pull samples toward the data manifold, suppressing off-manifold noise without retraining the model.
Standard diffusion sampling assumes clean training data and tends to reproduce training noise or artifacts. MAD fills this gap by providing an inference-time correction that enforces manifold consistency, enabling cleaner reconstructions from noisy-trained or imperfect models.
The ideas presented here are interesting conceptually, however the application of generation from noisy data training is not that useful in practice. Moreover, the experiments are mainly qualitative with not enough quantitative evaluations.

**Strengths:**

1.	Clear theoretical formulation. The “extended score” is well-defined and mathematically consistent.
2.	Simple, practical implementation. Requires only minor modifications to inference and reuses pretrained score networks.

**Weaknesses:**

1. The method is motivated by training on corrupted or noisy data, yet no controlled experiment on real data demonstrate this regime. As a result, the practical need and benefit of MAD are not motivated enough and it appears to fix a problem that the paper never clearly shows.
2. Image degradation at higher attraction strength. Increasing b reduces details and realism, questioning the method’s practicality for general image generation. Low b is equivalient to the regular score and as b increases image quality downgrades.
3. Minor related work references. The paper lacks a structured comparison to related inference-time or manifold-based diffusion approaches. Even if there is no method on this exact task, it would be beneficial to appropriately address other denoising methods. A comparison to some methods in the results section is also suggested (where denoising is illustrated, e.g. Fig 7).
4. No quantitative evaluation. Results remain purely qualitative, some emasures such as FID can be used to evaluate image realness.
5. Minor comment on figures - Figure 1 – plot titles should move to be y-axis labels. Ticks should be larger, it is very hard to read. Ticks are also very small in Figure 2, and values are not clear enough there.
6. No code is provided, hindering future comparisons.
7. The authors mention traditional manifold denoising methods but do not explain why they are irrelevant.
8. Persistent scanning artifacts or logos/watermarks seem like possible sources for noise that need removal in practice. The authors themselves mention compression artifacts. However, there is no examination of the presented method in such contexts.

**Questions:**

1. The paper claims to address training on corrupted or noisy data, but no controlled experiment on real data actually shows this. Can the authors provide concrete examples or benchmarks where this problem occurs in practice?
2. Increasing the manifold-attraction strength (b) consistently reduces image quality (Figs 3-5). Can the authors explain in what practical scenario the method should be used? Another question here – Section 5.2 and Figs 3-5 seem not relevant to the method (these datasets are noiseless as stated by the authors), why are they presented in the main body, and in such extent? Maybe removing or reducing the space of this part can leave some space for more practical use-cases.
3. Why is there no dedicated “Related Work” section comparing to inference-time denoising, posterior correction, or manifold-regularized diffusion methods?
4. Beyond visual comparisons, can the authors quantify improvement (e.g., denoising metrics, perceptual quality, FID) to justify the practical value of the approach?

---

> ### Author Response · Authors · 2025-11-21
>
> **Answers to weaknesses**
>
> 1. We added some controlled experiments with CIFAR-10 in section 5.4. Please see the general response, point 1.
>
>    The simulations with the EMPIAR-11618 dataset coming from Cryo-Electron Microscopy show that MAD can be successfully applied in a real world scenario. This dataset is intrinsically very noisy (Figure 8, top left), so that training can be done only on noisy data, and with MAD we are able to recognize all key features of these images. In this setting, it is not possible to carry out controlled experiments with both clean and noisy data, because clean data is unavailable.
>
> 2. The parameters $a$ and $b$ act as regularization parameters, and so these effects are natural and unavoidable: at one extreme, the denoising effect vanishes, and at the other extreme, key features of the images are treated as noise and tend to disappear. As in inverse problems, the key is finding suitable parameters. Please see the general response, point 3, for more details on this.
>
> 3. Please see the general response, point 2.
>
> 4. Please see the general response, point 1: we added FID scores.
>
> 5. Thank you for the comment, we will take care of it.
>
> 6. We are currently finalizing and cleaning the code. We are committed to releasing the full source code upon acceptance to facilitate future comparisons.
>
> 7. We are interested in the generative setting which is not considered by traditional manifold denoising methods. We have added a brief clarification of this in the beginning of Section 5. Also see point 2 of the general answer.
>
> 8. These are very interesting potential applications of our method. In this work, we focused on introducing and describing the method, discussing some experiments for illustrative purposes. The added simulations on pixel removal (see the general response, point 1) are an indication of the potential applicability to logos/watermarks removal. However,  extensive numerical simulations in different contexts are left for future work.
>
> **Answers to questions**
>
> 1. Please see the general response, point 1, and the discussion at the beginning of section 5. We added controlled experiments on real data, which complement the experiments with real data coming from Cryo-EM.
>
> 2. Please see the general response, points 3 and 4, and the discussion at the beginning of section 5. Increasing the parameter $b$ is useful when a very large noise is present. We moved Figure 5 to the Appendix.
>
> 3. Thank you for this suggestion. We chose to include related work in the Introduction to maintain a compact presentation. The reviewer is correct that we did not explicitly compare our work to "posterior correction" or other "inference-time" methods, and we added this context to the introduction in our revision.
>
> 4. Please see the general response, points 1: we added FID scores.

---

### Official Review · Reviewer_gCYR · 2025-10-31

**Soundness:** 2
**Presentation:** 3
**Contribution:** 2
**Rating:** 6
**Confidence:** 3

**Summary:**

This paper extends the standard score operator via
$
p \mapsto (1+\gamma) S(p * g_\gamma) + \frac{d}{d\gamma} S(p * g_\gamma),
$
where $S$ is the usual score and $g_\gamma$ is Gaussian smoothing. As $\gamma \to 0$, the extended score reduces to the standard score (when properly defined). This extension accommodates distributions without proper densities (e.g., multi-Dirac measures), partially addressing score blow-ups (Dirac/multi-Dirac scores scale like $1/\sigma$ and diverge).

The core intuition is that the extended score tends to **shrink low-variance directions**. For example, the standard score of an isotropic Gaussian coincides with the extended score of a Dirac. Sampling with the extended score may therefore reduce small variances that, under a manifold hypothesis, correspond to off-manifold noise/deviations in images. In this way, using the extended score implicitly enforces a manifold constraint during sampling—**Manifold Attracted Diffusion (MAD)**.

The extended score can be computed from a learned (standard) score. The authors leverage the EDM setup, select $\gamma(t)=\sigma(t)^p$ (with $\sigma(t)$ the noise level), and propose corresponding correction coefficients and sampler features. Visualizations on toy data, real image datasets (FFHQ, AFHQv2, and ImageNet), and scientific data show a variance-reduction/thresholding effect and suggest potential for producing cleaner images even when training data are corrupted.

**Strengths:**

1. The mathematical presentation is rigorous and instructive. Extending the predominant score operator is an interesting and timely idea.
2. The figures (esp. Figs. 1–2) are clear and intuitive. Experiments span synthetic, natural image, and scientific data, with seemingly stable and reasonable generations.

**Weaknesses:**

1. The justification for sampling with the extended score is not explicit. Beyond the Dirac case (line 216-225), the effects are mainly illustrated empirically, and the image-domain interpretations feel somewhat loose. The writing in these parts can be elusive.
2. Related to (1), the roles of the hyperparameters $a$ and $b$ are not explained intuitively. Empirically, increasing $a$ appears to reduce $\gamma(t)$, pulling the extended score closer to the standard one (weakening MAD), while $b$ affects both $\gamma$ and the correction term $m$, and in practice seems to strengthen MAD. This likely necessitates a hyperparameter search, which hurts applicability.
3. MAD roughly doubles computational overhead at inference.

**Questions:**

1. The sampler requires a derivative with respect to time/noise level. Some recent work shows the denoiser can sometimes omit explicit time conditioning [1]. Would MAD still be applicable in such settings, or does it fundamentally require time-dependent scores?
2. In Fig. 3 (rows 3–4), varying $b$ appears to modulate generation non-monotonically (e.g., a man → woman → man). Do the authors have an interpretation for this zig-zag behavior?

[1] Sun, Qiao, et al. *Is Noise Conditioning Necessary for Denoising Generative Models?* ICML 2025.

*My current rating is not a final assessment, as I am not an expert on diffusion model sampling. I may revise it after discussion with the authors and other reviewers.*

---

> ### Author Response · Authors · 2025-11-21
>
> **Answers to weaknesses**
>
> 1. We started with a mathematically grounded formulation, but the implementation required less clear choices.   This is  absolutely true, but unfortunately it is true also with the established sampling methods for diffusion models.
>
> 2. Please see the general response, point 3.
>
> 3. Please see the general response, point 5.
>
> **Answers to questions**
>
> 1. We thank you for the reference to Sun et al. (2025). Our current derivation relies on the derivative with respect to noise level $\gamma(t)$. It is an interesting open question whether ``manifold attraction'' can be achieved without explicit time-dependence, and we have highlighted this in the Conclusion section.
>
> 2. This is a sharp observation. We interpret this non-monotonic behavior as evidence of the complex, non-linear dynamics of the generative process, similar to varying the scale in Classifier-Free Guidance. A small change in $b$ can cause the sampling trajectory to cross a bifurcation point, leading the process to a different final attractor (e.g., a different face).

---

> ### Comment · Reviewer_gCYR · 2025-11-25
>
> Thanks for all the replies. I appreciate the case study in Appendix A.1 on the extended score for distributions with low-dimensional subspace support (where the extended score points to zero for the off-subspace entries, as in the Dirac case, quickly dragging them back to the subspace).
>
> 1. **(W1, W2)** Regarding the hyperparameter choice: although making an analogy with CFG is a reasonable argument, (1) CFG is arguably more useful and dominant in practice; (2) CFG also has a solid justification for the guidance strength [1, 2], while here the effect of $a, b$ is not entirely intuitive to me.
>
> 2. **(Q2)** I appreciate the authors’ honesty, but the non-monotonic generation behavior as $b$ changes may hurt the feasibility of MAD in practice.
>
> 3. The writing remains a bit elusive. I understand that the authors have their own writing style, but reading the paper still gives me the impression of focusing on an extension of the score operator, with applications to manifold attraction that appear somewhat incidental.
>
> That said, I lean towards maintaining my current positive rating.
>
> [1] Ho, Jonathan, and Tim Salimans. *Classifier-free diffusion guidance.* arXiv preprint, 2022.
> [2] Pavasovic, Krunoslav Lehman, et al. *Classifier-Free Guidance: From High-Dimensional Analysis to Generalized Guidance Forms.* arXiv preprint, 2025.

---

### Official Review · Reviewer_LVkS · 2025-10-31

**Soundness:** 2
**Presentation:** 2
**Contribution:** 2
**Rating:** 4
**Confidence:** 3

**Summary:**

This paper proposes MAD, an inference algorithm for diffusion generation under the setting where the diffusion model was trained on noisy data, but at inference time one wants to generate clean data. The proposed algorithm is based on the intuition that meaningful semantics lie on a lower dimensional manifold, while noise is orthogonal off the manifold. Based on this, at each denoising step, the proposed algorithm calculates the derivative of the score with respect to time, and updates the image using a weighted sum of score and the derivative. The paper provides motivational and intuitive analysis on synthetic settings and on applying the algorithm on diffusion models trained on clean real-world data, and performs denoising experiments on the intended setting on a synthetic dataset and a real dataset EMPIAR-11618.

**Strengths:**

* This paper looks at an interesting setting of obtaining clean generations from a diffusion model trained on noisy data, and proposes an inference only method that does not require retraining.
* The proposed method is technically sound and theoretically grounded.
* The illustrative examples and generations on FFHQ, AFHQV2, and ImageNet help gain intuition of the proposed algorithm's behavior.
* Qualitative visualizations on the synthetic data and the EMPIAR-11618 dataset look promising.

**Weaknesses:**

* The experimental evaluations are limited, relying primarily on qualitative visualizations. For the denoising experiments, it would be helpful to include quantitative metrics, such as FID scores computed against the clean data distribution.
* The two datasets in the denoising experiments have very simple semantics, making it hard to assess the scope of general applicability of the proposed method. It would be helpful to provide evaluations on more complex datasets,  e.g. CIFAR or ImageNet. Although the visualizations in 5.2 involve complex data, they are not a direct application of MAD’s intended setting but used diffusion models trained on clean data instead.
* The performance appears sensitive to the hyperparameters, and there is no obvious rule-of-thumb across different datasets. This could introduce non-trivial hyperparameter sweep overhead when applying MAD in practice, especially if, e.g. the dataset is complex and involve various sub-categories.
* It would be helpful to provide some evaluations and analysis on the inference speed and potential speed-performance tradeoffs. By default, each denoising step of MAD invokes two forward passes of the scoring network, which could raise concerns about significant computational overhead. It would be helpful to see, e.g. if MAD can work well with sampling speedup strategies (e.g. the simplest one could be using fewer sampling steps).

**Questions:**

* It would be helpful to see quantitative evaluations, such as FID with the clean data
* It would be helpful to see more discussion or analysis on the scope and applicability of the proposed method:
    * How well does it scale with dataset complexity, e.g. for datasets that go beyond simple semantics and strong inductive bias like CIFAR, ImageNet?
    * How does the performance look like when there are different levels of corruption/noise in the training data?
    * What kinds of corruption is the method effective against? Is it primarily designed for general noise, or does it also work for other corruption types such as blurring, compression artifacts, or other distortions? e.g. in the synthetic setting, would the method still be applicable if only the first step of blurry blobs was applied without the second step of adding Gaussian noise?
    * It would be useful to have some discussion or analysis on contextualizing the performance of this inference-time approach relative to training-time methods or finetuning with small amount of clean data methods. Is the proposed method's performance significantly worse, indicating a performance-compute tradeoff, or is it comparable? Additionally, a potential relevant inference-time baseline to consider for comparison is truncated sampling [1].

[1] Daras, Giannis, Yeshwanth Cherapanamjeri, and Constantinos Daskalakis. "How much is a noisy image worth? data scaling laws for ambient diffusion." ICLR 2025.

---

> ### Author Response · Authors · 2025-11-21
>
> **Answers to weaknesses**
>
> 1. Please see the general response, point 1: we added FID scores.
>
> 2. Please see the general response, point 1, on additional experiments, and point 4, on the relevance of the simulations with clean data.
>
> 3. Please see the general response, point 3.
>
> 4. Please see the general response, point 5.
>
> **Answers to questions**
>
> 1. Please see the general response, point 1: we added FID scores.
>
> 2.1. Please see the general response, point 1.
>
> 2.2. As expected, performance may get worse as the noise level increases. However, the simulations for the synthetic dataset (section 5.3) and for the real world data from Cryo-EM (section 5.5) show that the method can handle also very large levels of noise/corruption.
>
> 2.3 Regarding your question on specific corruptions:
>    * Gaussian bump functions without Gaussian noise: yes
>    * Blur: MAD currently struggles with Gaussian blur (on both synthetic and CIFAR-10 data). This is expected: this involves solving an inverse problem, which would need to be done with posterior sampling. Mathematically, the blurring modifies the manifold itself (essentially removing all low-frequency parts), and so the original (clean) manifold cannot be recovered with this method.
>    * Pixel Removal: It performs well on pixel removal (synthetic data, simulations added);  we are currently testing pixel removal, as well as box inpainting, on other datasets (results will be included in the last revision).
>    * Real Noise: The EMPIAR dataset represents complex, real-world Poisson-like noise, where MAD succeeds.
>
> 2.4. As noted in the General Response, point 2, we believe that a direct comparison is difficult due to the lack of established benchmarks for blind denoising where the degradation is unknown. However, we have added a discussion contextualizing MAD against truncated sampling and training-time methods in the Conclusion. We also added a comparison with truncated sampling (see section 5.4 and Figure 15 in the Appendix).

---

### Official Review · Reviewer_2prx · 2025-11-01

**Soundness:** 3
**Presentation:** 3
**Contribution:** 2
**Rating:** 4
**Confidence:** 3

**Summary:**

The paper proposes a novel sampling method for generating noiseless images from noisy training data. It mathematically defines an extended score that suppresses small variations while preserving large ones. This formulation motivates noiseless generation, as on-manifold directions typically exhibit large variations, whereas off-manifold directions correspond to small variations. Building on the extended score, the authors introduce a new sampling method and empirically evaluate it on both toy and real-world datasets.

**Strengths:**

The paper introduces an interesting new concept called the extended score. It provides mathematical proofs of several noteworthy properties of this term. Building on these properties, the authors propose a novel sampling algorithm and demonstrate its effectiveness through empirical experiments. Overall, the paper is well-motivated and clearly structured.

**Weaknesses:**

1. In Section 3, the paper proves the properties of the extended score under the Multi-Delta and Gaussian distributions with $\gamma = 0$, but then naturally extends these results to more complex real-world data distributions and $\gamma \neq 0$. This introduces a significant gap between theory and practice.

2. As shown in Figures 2–5, the proposed sampling method tends to pull the generated samples toward high-likelihood regions. In other words, although the method effectively reduces off-manifold noise, it also significantly alters the on-manifold distribution. Consequently, the noiseless sampling comes at the cost of reduced diversity. Also, these results are contradict to the statement "leave on-manifold variations unchanged".

3. For the noiseless generation experiments, the paper only presents results in Figures 7 and 8, which are relatively simple or toy examples. Moreover, the results are consistent with my concern above, that the generated samples exhibit very low diversity.

**Questions:**

1. In the abstract, the paper states: “The extended score … can be used to reduce small variations to zero, while leaving large variations mostly unchanged.” However, I find it unclear how this claim is theoretically justified in Section 3. In particular, Equation $H_0 \delta = S g_1$ (line 165) seems to indicate that the extended score of a delta distribution (variance → 0) is equal to the score of a Gaussian distribution with variance 1. In other words, it appears to increase the variance from 0 to 1, which seems contradictory to the statement that it “reduces small variations to zero.” Could the authors clarify this point further?

2. I would suggest evaluating the proposed method on more complex datasets such as CIFAR-10, FFHQ, or ImageNet. Additionally, when adding noise to images, it would be interesting to explore different types of noise, such as pixel removal or Gaussian blur, to better demonstrate the robustness of the proposed approach.

3. I also notice some conceptual similarities between classifier-free guidance and the sampling method proposed in this paper. Both approaches push the generation process toward high-likelihood regions. It may be worthwhile for the authors to explore how the proposed sampling method could potentially improve generation quality, perhaps offering a more interpretable "classier-free guidance".

---

> ### Author Response · Authors · 2025-11-21
>
> **Answers to weaknesses**
>
> 1. In Section 3, we analyze the Multi-Delta and Gaussian distributions at the limit $\gamma=0$ to derive precise analytical properties. This provides the theoretical intuition for the extended score. In practice, we cannot compute this limit and instead use sequences $\gamma(t) \to 0$. Such kinds of theory-practice gaps are common for diffusion models. For instance, precisely following the motivating diffusion process would require to start with a sample of $X_\sigma$ for some finite noise level depending on the training and going to $\sigma=0$; however, in practice, one starts with a fully Gaussian image, i.e. $\sigma=\infty$ and usually only goes to some small $\sigma>0$.
>
> 2. You are correct that MAD pulls samples toward high-likelihood regions, which inherently reduces diversity. This is the intended trade-off: we sacrifice some variance (both off-manifold noise and some on-manifold variation) to successfully recover clean structures from extremely noisy data—a task where standard sampling fails.
> We always acknowledge that the on-manifold variations are not perfectly preserved, but there is a trade-off. See also the general response, point 3.
>
> 3. Please see the general response, point 1: we added additional simulations, and included FID scores. We agree that, in particular, Figure 6 might suggest this; however, these are just a few example image from a toy data set with rather low diversity. As can be seen in the newly added Figure 7 for CIFAR-10, there does not seem to be any reduction in diversity in this case.
>
> **Answers to questions**
>
> 1. We thank you for pointing this out. The equation equates the extended score of a Delta to the score of a unit-variance Gaussian, but the practical implication goes in the opposite direction. In a noisy dataset, noise manifests as small, positive variance. The property $H_0\delta \approx Sg_1$ allows us to treat this low-variance noise as if it were a Delta distribution. Essentially, in case of variance (in a given direction) below some threshold, the sample is moved (in that direction) at least as fast as in the Dirac delta case.
>
>    Consequently, the inference process ``collapses'' this off-manifold variance toward zero. The parameters $a$ and $b$ control the sensitivity of this thresholding. We have clarified this interpretation in Sections 3 and 4.
>
> 2. Please see the general response, point 1: we added additional simulations, with more complex datasets and with different types of noise (pixel removal).
>
> 3. We thank the reviewer for this insightful connection. We agree that MAD could be viewed as an ``interpretable classifier-free guidance'', pushing generation toward the manifold. We have added this perspective to our conclusion as a direction for future work.

---

### Author Response · Authors · 2025-11-21
**General Response to All Reviewers**

We thank all reviewers for their constructive feedback and positive assessment of the core idea behind Manifold Attracted Diffusion (MAD). Since several reviewers raised similar points regarding quantitative evaluation, comparisons with baselines, hyperparameter sensitivity, the simulations with clean data and the computational cost, we address these common themes here.

1. **Quantitative Evaluation on Complex Datasets (Reviewers 2prx, LVkS, 4Etn)**. A common concern was the reliance on qualitative visual results. In response, we have significantly expanded our experimental section:

     * New Dataset: We added experiments on CIFAR-10 corrupted with additive Gaussian noise in the new Section 5.4.
     * Corruption Types: as suggested, we considered pixel removal, and added simulations  for the synthetic dataset, and are finalizing results for other datasets, which will be included in the last version of the revision. These results demonstrate that MAD can be effective beyond the Gaussian setting.
      * Quantitative Metrics: We computed FID scores for these new experiments, comparing MAD against standard sampling trained on the same noisy dataset. See the newly added Table 1.

2.  **Comparison with Baselines and Related Work (Reviewers LVkS, 4Etn).** Reviewers asked for comparisons with training-time methods, finetuning, or inference-time baselines like truncated sampling.
     * Scope of MAD: We emphasize that MAD addresses a specific blind generative denoising task where only noisy data is available, and the specific degradation model may be unknown. This distinguishes it from inverse problem solvers (like DPS) that require a known forward operator, or methods that require clean data for finetuning. Our work cannot be directly compared with standard approaches for manifold denoising, which are not generative, or with algorithms for denoising single images. We added a comment on this at the beginning of section 5.
     * Comparison: While a direct ``apples-to-apples'' benchmark for this specific setting is not standard, we agree that contextualizing the performance-compute trade-off is vital. We have expanded the Introduction and Conclusion to explicitly contrast MAD with posterior correction and truncated sampling. We also added Figure 15, in which we show that truncated sampling does not yield the desired denoising effect.

3. **Hyperparameters and Trade-offs (Reviewers gCYR, LVkS, 4Etn).** Reviewers noted the sensitivity to parameters $a$ and $b$. We agree that their role was not well explained in the first version.

     * Regularization Role: We view $a$ and $b$ as regularization parameters. As with any regularization (or inverse problem) technique, there is an inherent trade-off: at one extreme, the denoising effect vanishes; at the other, key semantic features may be treated as noise and erased. We added a comment on this after equation (4).
     * Optimization: As in regularization theory for inverse problems, the search for optimal regularization parameters typically represents a different line of research, and does not fit the scopes of this work. The current parameters allow users to control the "aggressiveness" of the manifold attraction based on their specific noise levels.

4. **Use of Pretrained Models on Clean Data (Section 5.2) (Reviewers 4Etn, LVkS).** Questions were raised about the relevance of Section 5.2 (FFHQ/ImageNet). These experiments are intended to build intuition by isolating the ``manifold attraction'' property in a controlled setting. We added a comment on this at the beginning of section 5. For space reasons, as suggested, we moved Figure 5 to the Appendix.

5. **Inference Speed and Compute (Reviewers LVkS, gCYR).** While the standard implementation doubles the forward passes, the computational overhead can be significantly reduced. In practice, the extended score is most critical in the final iterations of the reverse process to clean up the manifold structure. Applying MAD only in the later stages could easily reduce the computational overhead to only $\sim 50\%$ rather than $100\%$. We added a comment on that in the Conclusion. We also note that we have not looked into optimization of inference cost yet, and this can likely be further improved, e.g.\@ by only applying extended score to some of the inference steps but more strongly.

---

### Author Response · Authors · 2025-12-03
**Summary of revision (for the AC)**

Dear Area Chair,

All reviewers acknowledged the method's core value, describing the mathematical formulation as "rigorous and instructive" (gCYR), "technically sound" (LVkS), and "clear" (4Etn). The main criticism across reviewers was a reliance on qualitative visual results and a lack of complex baselines. These key concerns were thoroughly addressed in the revision. To assist your assessment, we summarize the main improvements below:

1. **Added More Datasets and Complex Degradation Models.** We included quantitative experiments on a noisy version of CIFAR-10 and we expanded beyond Gaussian noise to include pixel removal (Section 5.3). This complements the already present blind denoising of Cryo-EM data (Section 5.5), proving robustness against non-Gaussian, real-world corruptions.

2.  **Added Quantitative Metrics (FIDs).** We introduced Table 1 (Section 5.3 & 5.4), reporting FID scores on CIFAR-10 and synthetic data. MAD improves the FID on noisy CIFAR-10 from 147.47 (standard generation) to 77.15 (MAD), demonstrating quantitative superiority.

3. **Comparison with Baselines.** We explicitly compared MAD against Truncated Sampling (Reviewer LVkS’s suggestion). As shown in the newly added Figure 15, truncated sampling fails to remove noise patterns that MAD successfully filters out, validating that our method performs a distinct geometric operation.

4. **Ongoing Validation.** We are currently finalizing experiments on FFHQ under Gaussian noise, pixel removal, and Poisson noise. Preliminary results are consistent with the strong performance seen in our CIFAR-10 experiments, showing up to 65 % reduction in FID.

**Conclusion.** MAD proposes a novel inference-time method that leverages the manifold hypothesis to generate clean samples from noisy-trained models without requiring clean data for finetuning. We believe the revised paper fully addresses the initial reviews.

---

### Meta-Review · Area_Chair_JPmN · 2026-01-06

**Summary:**

This paper considers the setting where the data-generating distribution is given by adding noise to a manifold-supported distribution, and studies how diffusion models can be used to sample from the underlying distribution with low-dimensional support, effectively removing the observational noise. Reviewer all praised the idea of the paper as interesting, but complained about the experiments being overly qualitative, and the computational cost of inference being increased. I do not believe these main concerns were addressed during the rebuttal, and I thus recommend rejection.

**Reviewer Concerns:**

I believe concerns about hyperparameters and comparisons against other baselines were successfully addressed in the rebuttal. However, these are in my view relatively minor concerns:

- As for the cost of inference, the authors merely acknowledged the issue and hinted that it could be improved, but I believe that fair comparisons should always be made using the same number of function evaluations.

- Authors added FID numbers for noisy CIFAR-10 experiments. The fact that the authors had to add noise to CIFAR-10 to get the method to be useful highlights the concern raised by reviewer 4Etn: it is unclear how useful the method is in practice.

**Reviewer Scores:**

I believe reviewers would have all kept their scores.

---

### Decision · Program_Chairs · 2026-01-26

Reject